# Neuropathy-causing TRPV4 mutations disrupt TRPV4-RhoA interactions and impair neurite extension

Brett A. McCray ⬡ [1✉], Erika Diehl[2,3], Jeremy M. Sullivan[1], William H. Aisenberg[1], Nicholas W. Zaccor ⬡ [1], Alexander R. Lau[1], Dominick J. Rich ⬡ [1], Benedikt Goretzki ⬡ [2,3], Ute A. Hellmich[2,3,5], Thomas E. Lloyd ⬡ [1,4] & Charlotte J. Sumner ⬡ [1,4✉]

TRPV4 is a cell surface-expressed calcium-permeable cation channel that mediates cell-specific effects on cellular morphology and function. Dominant missense mutations of TRPV4 cause distinct, tissue-specific diseases, but the pathogenic mechanisms are unknown. Mutations causing peripheral neuropathy localize to the intracellular N-terminal domain whereas skeletal dysplasia mutations are in multiple domains. Using an unbiased screen, we identified the cytoskeletal remodeling GTPase RhoA as a TRPV4 interactor. TRPV4-RhoA binding occurs via the TRPV4 N-terminal domain, resulting in suppression of TRPV4 channel activity, inhibition of RhoA activation, and extension of neurites in vitro. Neuropathy but not skeletal dysplasia mutations disrupt TRPV4-RhoA binding and cytoskeletal outgrowth. However, inhibition of RhoA restores neurite length in vitro and in a fly model of TRPV4 neuropathy. Together these results identify RhoA as a critical mediator of TRPV4-induced cell structure changes and suggest that disruption of TRPV4-RhoA binding may contribute to tissue-specific toxicity of TRPV4 neuropathy mutations.

[1] Department of Neurology, Johns Hopkins University School of Medicine, Baltimore, MD, USA. [2] Department of Chemistry, Biochemistry Section, Johannes Gutenberg-Universität Mainz, Mainz, Germany. [3] Center for Biomolecular Magnetic Resonance (BMRZ), Goethe-Universität, Frankfurt am Main, Germany. [4] The Solomon H. Snyder Department of Neuroscience, Johns Hopkins University School of Medicine, Baltimore, MD, USA. [5] Present address: Institute of Organic Chemistry and Macromolecular Chemistry, Cluster of Excellence 'Balance of the Microverse', Friedrich-Schiller-Universität, Jena, Germany. ✉email: bmccray3@jhmi.edu; csumner1@jhmi.edu

A common enigmatic finding across diverse hereditary diseases is the selective vulnerability of specific cell types despite wide expression of disease genes beyond affected tissues and organs. In nearly all cases, the basis for tissue-specific pathology is unknown. Transient receptor potential vanilloid 4 (TRPV4)-related diseases provide a striking example of tissue-specific pathology. Whereas some mutations cause autosomal dominant Charcot–Marie–Tooth disease type 2C and related forms of spinal muscular atrophy[1–3], conditions characterized by variably severe motor axonal degeneration with resultant weakness of the upper and lower extremities, other distinct mutations cause unrelated diseases of bone and connective tissue, including various forms of skeletal dysplasia, inherited osteoarthropathy, and hereditary osteonecrosis[4–7]. There are only rare examples of patients manifesting combined phenotypes[8–10]. Thus, TRPV4 mutations cause a spectrum of human diseases affecting the nervous system, connective tissue, and bone, tissues that are linked by their requirement for a robust cytoskeleton that can adapt and respond to dynamic extracellular stimuli.

TRPV4 is a member of the TRP family of diverse cation channels that coordinate a variety of cellular processes by transducing environmental stimuli into downstream signaling events[11]. As a broadly expressed and plasma membrane-localized channel, TRPV4 can respond to a range of mechanical and chemical stimuli arising from both the extracellular space and cytoplasm[12–14]. TRPV4 functions as a homotetrameric ion channel with each subunit containing six transmembrane domains and a large cytosolic N-terminal domain[12]. The N terminus contains a subdomain known as the ankyrin repeat domain (ARD), which is comprised of six ankyrin repeats and is thought to be an important mediator of protein–protein interactions[4,12,15–17]. As with many other TRP channels, TRPV4 can modulate cellular morphology through regulation of cytoskeleton dynamics. TRPV4 exerts an influence on the microtubule and actin cytoskeleton, thereby regulating cell stiffness, migratory behavior, morphogenesis, and cellular process extension and retraction[18–23]. The TRPV4 ARD is thought to serve a scaffolding function that is particularly important for mediating the effects of TRPV4 on cytoskeletal changes[13,17]. However, there is limited information regarding specific protein–protein interactions occurring via the ARD, and the mechanistic details of how TRPV4 exerts such profound effects on cellular morphology are not well characterized.

The pathogenic mechanisms underlying TRPV4 neuropathy and skeletal dysplasia remain unknown. Studies of TRPV4 disease mutants in heterologous systems have demonstrated that both neuropathy and skeletal dysplasia mutations lead to gain of ion channel function, increased basal and stimulated TRPV4-mediated calcium influx, and cytotoxicity that can be rescued with channel antagonists[1,2,7,16,24,25]. Furthermore, neuronal degenerative phenotypes in a fly model of TRPV4 neuropathy are suppressed by either an inactivating mutation within the ion channel pore or treatment with a TRPV4 channel antagonist[26]. However, increased ion channel activity cannot solely account for the tissue-selective pathology caused by TRPV4 mutations. A potential clue to tissue specificity in TRPV4 disease comes from the observation that neuropathy mutations most frequently occur at highly conserved, surface-exposed arginine residues within the ARD[17]. In contrast, skeletal dysplasia mutations are found throughout multiple domains of TRPV4 in regions that may be critical for regulation of ion channel gating[4,27]. The clustering of neuropathy mutations within the TRPV4 ARD suggests that these mutations may specifically disrupt scaffolding functions of TRPV4. While there have been prior efforts to characterize the TRPV4 interactome[28–30], no studies have uncovered altered protein–protein interactions among various TRPV4 disease mutants.

In this study, utilizing an unbiased proteomics approach to identify TRPV4 interactors, we identified the small GTPase RhoA as a direct TRPV4-interacting protein. RhoA is a master regulator of cytoskeletal dynamics that acts as a molecular switch by cycling between an inactive GDP-bound state and an active GTP-bound state[31]. In its active state, RhoA favors actin polymerization, stress fiber formation, and actomyosin contraction, which generally result in cell contraction and reduced cellular process extension[32–34]. Here, we demonstrate robust reciprocal regulation of RhoA and TRPV4 function with TRPV4 exerting both ion channel-dependent and ion channel-independent modulation of RhoA activation and cytoskeletal changes. Remarkably, neuropathy mutations, but not skeletal dysplasia mutations, disrupt TRPV4–RhoA interactions and TRPV4-mediated regulation of cellular outgrowth. Together, our results demonstrate that neuropathy mutations within the TRPV4 ARD specifically disrupt the ion channel-independent scaffolding function of TRPV4, which may be an important pathological event contributing to neuron-specific disease.

## Results

**TRPV4 interacts with the small GTPase RhoA.** In ongoing efforts to identify novel TRPV4 protein–protein interactions that are relevant to TRPV4-related neuropathy, we used an unbiased proteomics approach in which we expressed C-terminal FLAG-tagged wild type (WT) or neuropathy mutant (R237L) TRPV4 in HEK293T cells followed by immunoprecipitation and liquid chromatography-mass spectrometry. Stratification of potential interactors based on relative enrichment in WT TRPV4 vs. R237L TRPV4 and negative control (empty vector transfection) identified several putative TRPV4 interactors (Supplementary Table 1). We chose to focus subsequent studies on the small GTPase RhoA as it demonstrated high spectral counts, appeared enriched with WT TRPV4, and plays a well-known role in regulating the actin cytoskeleton. In addition, RhoA was also recently identified as a putative TRPV4 interactor in a yeast two-hybrid screen[28]. We confirmed TRPV4–RhoA interaction by co-immunoprecipitation of epitope-tagged TRPV4 and RhoA from transfected MN-1 cells (Fig. 1a, b), a mouse motor neuron–neuroblastoma fusion cell line[35–37]. We also generated a stable HEK293T cell line that inducibly expresses FLAG-tagged WT or R269C neuropathy mutant TRPV4 in response to tetracycline treatment. These cells, hereafter referred to as T-Rex-TRPV4^WT or T-Rex-TRPV4^R269C cells, demonstrate time-dependent TRPV4 expression with appropriate complex glycosylation, cell surface localization, neuropathy mutation-dependent elevation of baseline calcium (Supplementary Fig. 1a–d), and only modest overexpression with 15 ng/ml tetracycline (Supplementary Fig. 1e). In T-Rex-TRPV4^WT cells, endogenous RhoA co-immunoprecipitated with TRPV4-FLAG, but not with nonspecific antibody (Fig. 1c), and not in the absence of TRPV4-FLAG induction (Supplementary Fig. 1f). We also used mouse choroid plexus to show that endogenous TRPV4 co-immunoprecipitated endogenous RhoA (Supplementary Fig. 1g). Rac1 and Cdc42, related Rho GTPases, did not co-immunoprecipitate TRPV4 in MN-1 cells, demonstrating specificity of the TRPV4–RhoA interaction (Supplementary Fig. 1h). Consistent with prior studies, transfected RhoA was predominantly diffusely localized within cells without enrichment at its target membranes[38]. However, we detected partial co-localization of transfected RhoA and TRPV4 within MN-1 cells (Fig. 1d) and COS7 cells (Supplementary Fig. 1i), both at the plasma membrane and within intracellular compartments. To determine whether TRPV4–RhoA interactions are direct, we generated immunopurified TRPV4 and RhoA from separate HEK293T cell lysates for use in a pull-down assay (see schematic

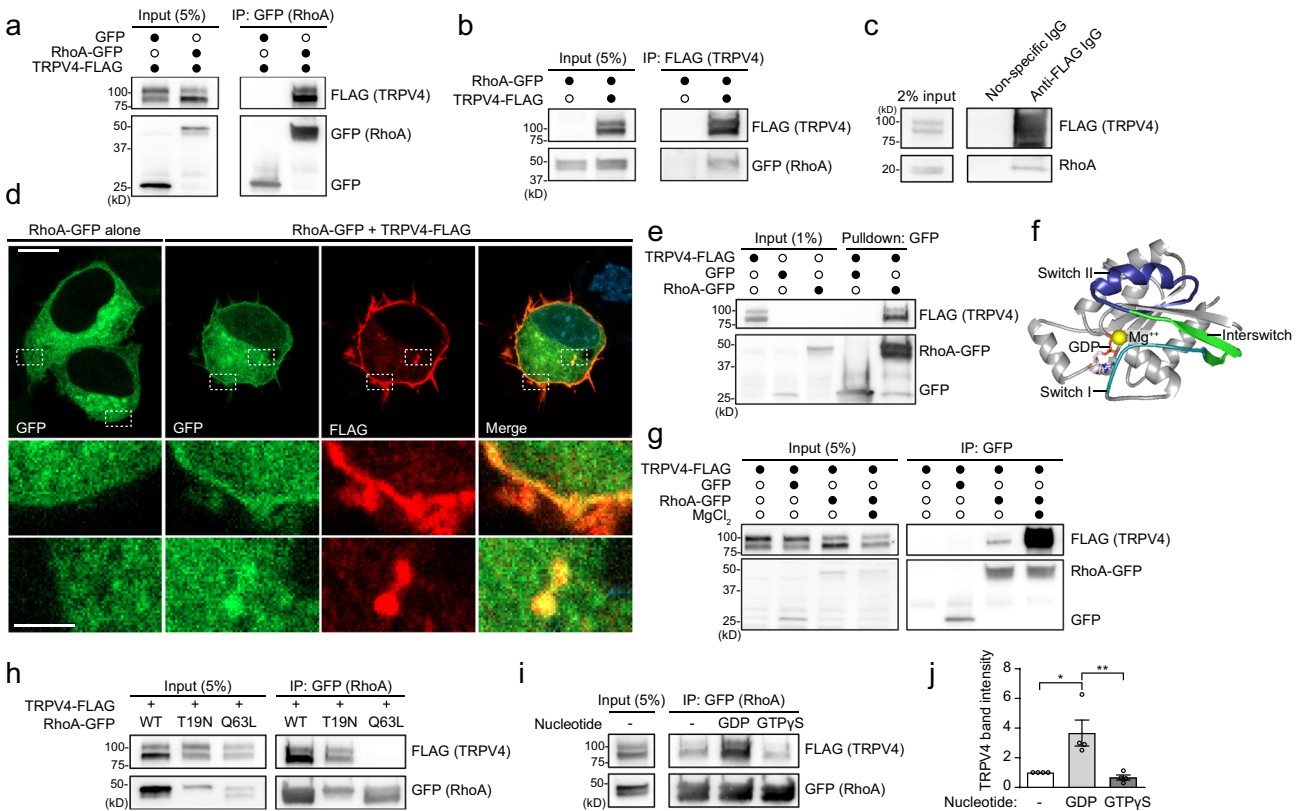

**Fig. 1 TRPV4 interacts with GDP-bound RhoA. a**, **b** MN-1 cells transfected with TRPV4-FLAG and GFP or RhoA-GFP were subjected to immunoprecipitation with anti-GFP (**a**) or anti-FLAG (**b**) antibody. **c** Stable, inducible TRPV4-FLAG-expressing HEK293T (T-Rex-TRPV4 cells) were treated with tetracycline (15 ng/ml) to induce TRPV4-FLAG expression and subjected to immunoprecipitation with anti-FLAG antibody or nonspecific antibody. Endogenous RhoA co-immunoprecipitates with TRPV4. **d** Immunofluorescence images of MN-1 cells transfected with TRPV4-FLAG and RhoA-GFP demonstrate co-localization of TRPV4 and RhoA at the plasma membrane and within intracellular vesicular structures. Scale bars, 10 and 2.5 μm for inset. **e** TRPV4-FLAG, RhoA-GFP, and GFP were expressed individually in HEK293T cells and subjected to immunoprecipitation with FLAG or GFP antibody followed by extensive washing. TRPV4-FLAG was eluted with FLAG peptide and then co-incubated with immunopurified RhoA-GFP or GFP. Immunopurified TRPV4 interacts with immunopurified RhoA, suggesting direct interaction. **f** Structural model of GDP-bound RhoA (PDB: 1FTN) showing the effector-interacting switch I (teal) and II regions (blue), interswitch region (green), magnesium ion (yellow) within the nucleotide-binding pocket, and GDP. **g** Co-immunoprecipitation of MN-1 cells transfected with TRPV4-FLAG and RhoA-GFP in the presence of EDTA (1 mM in all conditions), which disrupts RhoA nucleotide binding, with or without excess magnesium (10 mM), which stabilizes RhoA nucleotide binding. TRPV4–RhoA interactions are nucleotide dependent. **h** Co-immunoprecipitation of MN-1 cells transfected with TRPV4-FLAG and WT RhoA-GFP, dominant negative, inactive RhoA-GFP (T19N), or constitutively active RhoA-GFP (Q63L) demonstrates that TRPV4 interacts with WT RhoA and inactive RhoA T19N, but not active RhoA-Q63L. **i** Co-immunoprecipitation of MN-1 cells transfected with TRPV4-FLAG and RhoA-GFP with excess GDP or unhydrolyzable GTP (GTPγS) demonstrates more efficient interaction between TRPV4 and RhoA in the presence of GDP. **j** Densitometry of immunoprecipitated TRPV4 band intensity in the presence or absence of GDP or GTPγS; representative blot shown in **i**. One-way ANOVA with Dunnett's post hoc test, n = 4 independent experiments, *p = 0.013, **p = 0.0066. Data are presented as means ± SEM. Further details regarding the number of times experiments were repeated are presented in "Methods" under "Statistics and reproducibility".

of experimental design in Supplementary Fig. 1j). Co-incubation of immunopurified RhoA-GFP (Supplementary Fig. 1k) with purified TRPV4-FLAG (Supplementary Fig. 1k) led to robust pull-down, indicating direct interaction of TRPV4 and RhoA (Fig. 1e). Together, these data demonstrate that RhoA is a specific TRPV4-interacting protein.

**TRPV4 interacts with inactive, GDP-bound RhoA.** The activity cycle of RhoA is governed by guanine nucleotide exchange factors (GEFs) and GTPase activating proteins, which facilitate RhoA activation and inactivation, respectively[31]. In the activated GTP-bound state, the effector binding regions of RhoA, known as switch loops I and II, undergo conformational changes that facilitate interaction with downstream effectors[31]. RhoA nucleotide affinities are also strongly influenced by coordination of magnesium within the nucleotide-binding pocket (Fig. 1f). We

found that TRPV4–RhoA interaction was markedly enhanced by addition of magnesium to buffers that contain EDTA (Fig. 1g), suggesting that efficient TRPV4–RhoA interaction depends on nucleotide binding. To test whether TRPV4 interacts with GDP- or GTP-bound RhoA, we first used two well-characterized RhoA mutants: T19N, which abolishes the ability to bind GTP and keeps RhoA in an inactive, predominantly GDP-bound state to create a dominant negative, and Q63L, which disrupts GTP hydrolysis to produce GTP-locked, constitutively active RhoA[31]. In MN-1 cell lysates, TRPV4 co-immunoprecipitated with WT and T19N RhoA, but this interaction was abolished by the Q63L RhoA mutation (Fig. 1h), suggesting that TRPV4 preferentially binds GDP-bound RhoA. Interaction of TRPV4 with GDP-bound RhoA was further confirmed by co-immunoprecipitation in the presence of either excess GDP or an unhydrolyzable GTP analog (GTPγS) to favor the inactive or active form of RhoA, respectively. Whereas the TRPV4–RhoA interaction was markedly

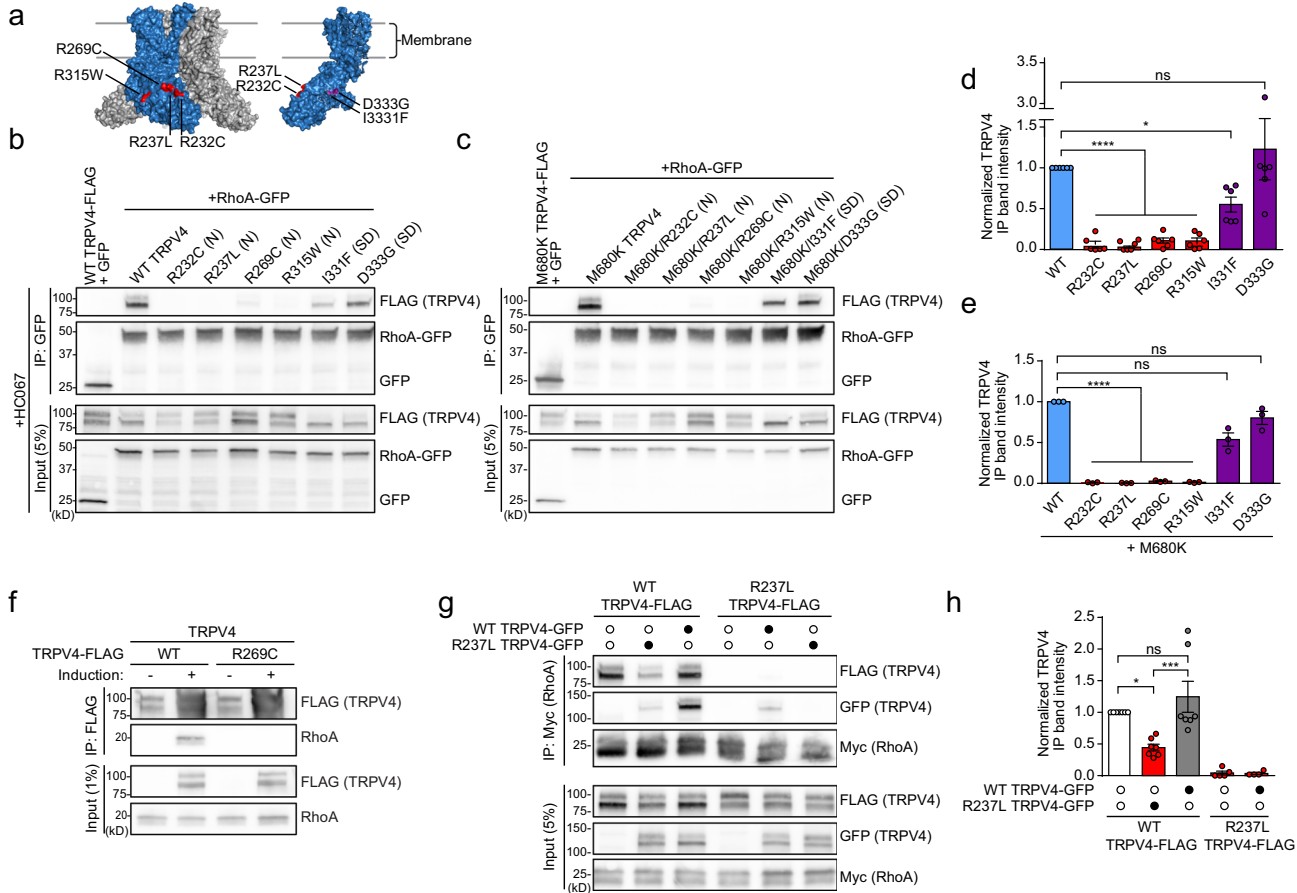

**Fig. 2 TRPV4 neuropathy mutations disrupt RhoA interactions. a** Schematic showing the structure of the TRPV4 homotetramer (*Xenopus tropicalis*, PDB: 6BBJ) with a single subunit in blue. Neuropathy mutations (R232C, R237L, R269C, R315W) are highlighted in red, and skeletal dysplasia mutations (I331F, D333G) within the opposite face of the ARD (rotated for visualization) are highlighted in purple. **b**, **c** MN-1 cells were transfected with RhoA-GFP and TRPV4-FLAG with neuropathy-causing mutations (N) or skeletal dysplasia-causing mutations (SD) and treated with the TRPV4 antagonist HC067 (0.5 μM) (**b**) or RhoA-GFP and TRPV4-FLAG harboring an ion channel pore-inactivating mutation, M680K, either alone or in combination with neuropathy-causing mutations (N) or skeletal dysplasia-causing mutations (SD) (**c**). Cell lysates were then subjected to immunoprecipitation with anti-GFP antibody. Neuropathy mutations, but not skeletal dysplasia mutations, disrupt interaction between TRPV4 and RhoA, independent of calcium-mediated cytotoxicity. **d**, **e** Quantification of densitometry of immunoprecipitated TRPV4 band intensity divided by TRPV4 input band intensity, normalized to WT TRPV4 ((**d**); representative blot shown in **b**) or M680K TRPV4 ((**e**); representative blot shown in **c**). One-way ANOVA with Tukey post hoc test, $n = 6$ (**b**) and $n = 3$ (**c**) independent experiments, *$p = 0.035$, ****$p < 0.0001$. **f** Co-immunoprecipitation using T-Rex-TRPV4$^{WT}$ or T-Rex-TRPV4$^{R269C}$ cells demonstrates that WT but not neuropathy mutant TRPV4 interacts with endogenous RhoA. The presence of TRPV4-FLAG in uninduced immunoprecipitation lanes is due to low-level expression leak. **g** Co-immunoprecipitation of MN-1 cells transfected with TRPV4-FLAG and TRPV4-GFP constructs as well as RhoA-GFP demonstrates that neuropathy mutant TRPV4 reduces interaction of WT TRPV4 with RhoA (compare lane 1 vs. 2). **h** Densitometry of immunoprecipitated TRPV4 band intensity normalized to WT TRPV4 alone (lane 1); representative blot shown in **g**. One-way ANOVA followed by Tukey's multiple comparisons test, $n = 4$ (lane 5), 5 (lane 4), or 7 (lanes 1, 2, and 3) independent experiments, *$p = 0.028$, ***$p = 0.0009$. Data are presented as mean ± SEM. Further details regarding the number of times experiments were repeated are presented in "Methods" under "Statistics and reproducibility".

increased with excess GDP, the interaction was reduced with excess GTPγS (Fig. 1i, j). These data indicate that TRPV4 interacts primarily with the inactive, GDP-bound conformation of RhoA, suggesting that interaction occurs through structural determinants in the RhoA effector binding switch I, switch II, and interswitch regions (Fig. 1f).

**TRPV4 neuropathy mutations disrupt RhoA interaction.** Given that the majority of neuropathy-causing mutations in TRPV4 cluster within the exposed face of the intracellular ARD that mediates protein–protein interactions, we tested whether neuropathy mutations specifically alter TRPV4–RhoA interaction. For these experiments, we examined four neuropathy mutants that affect conserved arginines within the convex surface of the ARD

(R232C, R237L, R269C, R315W) and two skeletal dysplasia mutants that are located within the opposite face of the ARD (I331F, D333G), which is less surface-exposed (Fig. 2a)[2,16,17,27]. To determine whether TRPV4–RhoA interaction is influenced by disease mutations, we performed co-immunoprecipitation from MN-1 cells transfected with tagged RhoA along with either WT or mutant TRPV4. Remarkably, all tested neuropathy mutations resulted in nearly complete disruption of TRPV4–RhoA interaction, whereas the skeletal dysplasia mutations displayed variably preserved interaction with RhoA (Fig. 2b–e and Supplementary Fig. 2a). In contrast, the R237L neuropathy mutant showed preserved interaction with PACSIN1, which interacts with the N-terminal proline-rich domain (PRD) of TRPV4 (Supplementary Fig. 2b), consistent with prior results[2]. As expression of neuropathy mutants can lead to calcium-

mediated cytotoxicity and cell death[2,16], and TRPV4 ion channel activity could potentially influence protein–protein interactions, we performed TRPV4–RhoA co-immunoprecipitation experiments in the presence (Fig. 2b–e) or absence (Supplementary Fig. 2a) of TRPV4 ion channel inhibition. Neither pharmacologic inhibition of calcium influx with the specific TRPV4 inhibitor HC067047 (HC067) (Fig. 2b, d) nor a pore-inactivating mutation (M680K)[39] (Fig. 2c, e) could restore TRPV4–RhoA binding in neuropathy mutants, indicating that disruption of TRPV4–RhoA interactions with neuropathy mutations occurs independent of ion channel activity. We also utilized the T-Rex-TRPV4 cells to demonstrate that the neuropathy mutant R269C disrupts interaction with endogenous RhoA (Fig. 2f). Consistent with these results, co-localization of tagged RhoA with TRPV4 in MN-1 cells and COS7 cells was also reduced with neuropathy mutant TRPV4, but not with skeletal dysplasia mutant TRPV4 (Supplementary Fig. 2c, d).

TRPV4 channels are composed of four TRPV4 subunits that assemble into homotetramers[12]. Given that TRPV4 neuropathy mutations cause autosomal dominant disease and cause a gain of function in vivo[26], we hypothesized that the co-expression of mutant TRPV4 might disrupt the normal binding of WT TRPV4 with RhoA. We first tagged WT and neuropathy mutant TRPV4 with different epitope tags and used co-immunoprecipitation to confirm that WT and mutant TRPV4 were able to associate in cells (Supplementary Fig. 2e). We next performed co-immunoprecipitation with over-expressed RhoA and FLAG-tagged WT TRPV4 alone or in combination with GFP-tagged WT or neuropathy mutant TRPV4. We found no significant difference in the co-immunoprecipitation of FLAG-tagged WT TRPV4 with RhoA when GFP-tagged WT TRPV4 was co-expressed, but we found a striking reduction in the co-immunoprecipitation of WT TRPV4 with RhoA when GFP-tagged mutant TRPV4 was co-expressed (Fig. 2g, h). These results suggest that incorporation of mutant TRPV4 into the TRPV4 tetramer suppressed interaction of WT TRPV4 with RhoA, consistent with an inhibitory effect of mutant TRPV4 on RhoA interaction.

**The TRPV4 ARD directly binds RhoA**. We next sought to identify the region of TRPV4 that interacts with RhoA. Given that neuropathy mutations within the convex face of the N-terminal ARD of TRPV4 disrupt RhoA interaction, we examined the interaction of RhoA with the TRPV4 N terminus. We generated and purified recombinant human TRPV4 N-terminal domain (hV4-NTD), which could be recognized by an antibody directed against the proximal TRPV4 N terminus (Supplementary Fig. 3a). We also generated recombinant TRPV4 ARD WT (hV4-ARD-WT), TRPV4 ARD R269C neuropathy mutant (hV4-ARD-R269C), and full-length RhoA (Supplementary Fig. 3b). Molecular weights and secondary structure of recombinant proteins as determined by circular dichroism (CD) spectroscopy were as expected (Supplementary Fig. 3c, d)[17,40]. Additionally, purified RhoA protein was functional in a luminescence-based GTPase assay (Supplementary Fig. 3e). Using immunopurified RhoA-Myc from HEK293T cells, we found specific pull-down of hV4-NTD with WT RhoA-Myc, but not constitutively active, GTP-bound RhoA-Q63L-Myc (Fig. 3a). Next, using hV4-ARD-WT and cell lysates from cells expressing RhoA-GFP, we demonstrated specific RhoA-GFP pull-down by the isolated human TRPV4 ARD (hV4-ARD), demonstrating direct interaction of the TRPV4 ARD with RhoA (Fig. 3b). We also interrogated whether domains in the TRPV4 N terminus outside of the ARD could influence RhoA interaction. We examined a PI(4,5)P$_2$ binding domain mutant ($^{121}$AAWAA$^{125}$) that alters channel activity and function[40,41] as

well as a PRD mutant (P142A/P143L), which has been shown to interfere with PACSIN interaction[29]. Neither of these mutants caused significant disruption of RhoA binding (Supplementary Fig. 3f), although it remains possible that regions outside the TRPV4 ARD can modulate RhoA interaction.

To characterize the interaction of TRPV4 ARD and RhoA in more detail, we recorded two-dimensional $^1$H, $^{15}$N nuclear magnetic resonance (NMR) spectra of $^{15}$N-isotope labeled human GDP-bound RhoA (hereafter referred to as RhoA) (Supplementary Fig. 3h). The spectral dispersion of the NMR spectra indicated that isotope-labeled RhoA was well folded. In addition, the spectral agreement with previously published NMR data (BMRB: 16668)[42] allowed the transfer of 93.6% of the backbone resonance assignments as a prerequisite for mapping the TRPV4 ARD binding site on RhoA. We then recorded spectra of RhoA in the presence of either hV4-ARD-WT or hV4-ARD-R269C (Supplementary Fig. 3h, i). With hV4-ARD-WT, we observed decreases in signal intensities for RhoA resonances, indicating formation of a large molecular complex between hV4-ARD-WT and RhoA (Fig. 3c, d and Supplementary Fig. 3j). In contrast, addition of hV4-ARD-R269C did not lead to a strong decrease in the signal intensities of the RhoA NMR spectra (Fig. 3c, d and Supplementary Fig. 3i), indicating a reduced affinity for RhoA, consistent with our co-immunoprecipitation results. To identify the RhoA residues that are strongly affected by the interaction with hV4-ARD-WT, we determined the relative signal intensity for every residue in the RhoA NMR spectra. In total, peaks corresponding to 20 residues in $^{15}$N-RhoA exhibited peak integral deceases greater than two standard deviations (2σ) in the presence of hV4-ARD-WT (Supplementary Fig. 3j), whereas only two residues showed this decrease with hV4-ARD-R269C. This is also reflected in the mean relative signal intensity change of RhoA in the presence of hV4-ARD-WT vs. hV4-ARD-R269C (Fig. 3d). Mapping the affected residues onto the crystal structure of RhoA (PDB: 1FTN)[43] revealed that the interswitch and switch II regions of RhoA are primarily involved in TRPV4 ARD binding (Fig. 3e). We noted the largest decrease in signal intensity for the E54 residue in RhoA, and a smaller decrease in E47, suggesting these residues might be particularly important in TRPV4 binding. Indeed, introducing an E54A mutation into RhoA completely disrupted interaction with TRPV4, whereas interaction was preserved with the E47A mutant (Fig. 3f). Both mutants showed preserved interaction with RhoGDI, indicating that they were properly folded and functional (Supplementary Fig. 3g).

**RhoA inhibits TRPV4 ion channel function**. The TRPV4 N terminus is important for channel activity and interaction with regulatory protein partners[29,40,41,44,45], raising the possibility that RhoA binding to the ARD could influence TRPV4-mediated calcium influx. As RhoA plays a critical role in regulating the actin cytoskeleton, actomyosin contraction, and cell morphology, we hypothesized that TRPV4–RhoA interactions may be particularly relevant in the context of TRPV4 response to the mechanical stimulus of hypotonic stress[12]. We first established that hypotonic stress in MN-1 cells does not lead to increased calcium influx in the absence of exogenous TRPV4 expression (Supplementary Fig. 4a), thus indicating that there are few if any endogenous osmotically activated calcium channels present in these cells. Upon expression of WT TRPV4-GFP, MN-1 cells displayed a mildly increased basal calcium level and a robust increase in intracellular calcium following treatment with hypotonic saline (Fig. 4a–c and Supplementary Fig. 4b). Co-expression of RhoA caused a marked inhibition of TRPV4-mediated calcium influx. Similarly, pharmacological activation of TRPV4 with the

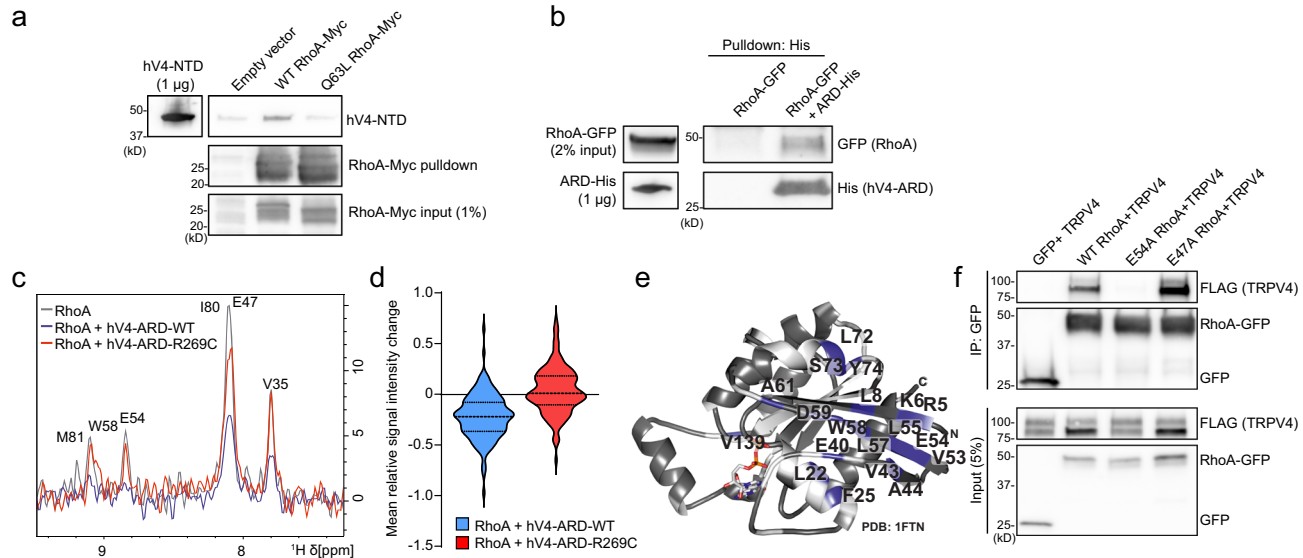

**Fig. 3 The TRPV4 ARD is sufficient for RhoA binding. a** RhoA-WT-Myc or RhoA-Q63L-Myc were immunopurified from HEK293T cells and then incubated with 10 μg purified human TRPV4 N-terminal domain (hV4-NTD). hV4-NTD is pulled down by WT RhoA, but not RhoA-Q63L. **b** Pull-down of lysate from cells expressing RhoA-GFP with purified human His-tagged TRPV4 ARD (hV4-ARD-His) bound to NiNTA beads demonstrates direct interaction of RhoA with the TRPV4 ARD. **c** Representative 1D projection of $^1H$ resonances from 2D $^1H$–$^{15}N$ NMR spectra of $^{15}N$-RhoA (gray) and in the presence of either hV4-ARD-WT (blue) or hV4-ARD-R269C (red). Stronger signal decreases for RhoA in the presence of WT ARD vs. neuropathy mutant indicate decreased interaction between RhoA and mutant ARD. **d** Violin plot showing integral changes of the 145 assigned backbone amide RhoA resonances in the presence of unlabeled hV4-ARD-WT or hV4-ARD-R269C. Greater average signal intensity changes in the RhoA spectra with hV4-ARD-WT are indicative of increased complex formation with WT as compared to the neuropathy mutant. Dashed lines represent means, dotted lines represent quartiles. **e** NMR signal intensity changes in the presence of WT TRPV4 ARD were mapped onto the crystal structure of GDP-bound human RhoA (PDB: 1FTN)[92]. Residues with significant decreases in signal intensity (labeled with residue position and indicated in blue) are predominantly localized in the RhoA switch regions. White portions correspond to residues for which no peaks could be identified in the measured $^1H$–$^{15}N$ 2D HSQC spectra. **f** Co-immunoprecipitation of MN-1 cells transfected with TRPV4-FLAG and WT RhoA-GFP, E54A RhoA-GFP, or E47A RhoA-GFP demonstrates disruption of TRPV4–RhoA interaction with mutation of the E54 residue of RhoA. Further details regarding the number of times experiments were repeated are presented in "Methods" under "Statistics and reproducibility".

specific TRPV4 agonist GSK1016790A (GSK101) led to calcium influx that was attenuated by co-expression of RhoA (Supplementary Fig. 4c, d). These effects were not due to altered TRPV4 surface expression in the presence or absence of RhoA as assessed by cell surface biotinylation (Supplementary Fig. 4e).

Given that overexpression of RhoA leads to actomyosin contraction, formation of stress fibers, and a rounded cellular morphology[32] (see also Fig. 4a, compare TRPV4-GFP images on left), we investigated whether these morphological changes on their own could be responsible for the reduced TRPV4 response to hypotonicity, or whether TRPV4–RhoA interaction was necessary for the inhibitory effect. To differentiate between these possibilities, we expressed constitutively active RhoA (Q63L), which does not bind TRPV4 (Fig. 1h) but leads to the same cellular morphological changes as expression of WT RhoA (Fig. 4a, compare TRPV4-GFP images on left). TRPV4 responses to hypotonic saline with constitutively active RhoA were indistinguishable from conditions without overexpression of RhoA (Fig. 4a–c), suggesting that inhibition of TRPV4 channel activity is driven by TRPV4–RhoA interaction rather than an indirect effect of changes in cell morphology. We next addressed whether RhoA could suppress calcium influx via neuropathy mutant TRPV4. Expression of mutant TRPV4 caused marked elevation of baseline calcium levels (Fig. 4a and Supplementary Fig. 4b), in agreement with previous studies[2,16]. Somewhat surprisingly, RhoA expression suppressed baseline calcium levels with expression of mutant TRPV4 (Fig. 4a and Supplementary Fig. 4b), which may reflect a nonspecific effect of altered cytoskeletal dynamics with overexpression of RhoA, or perhaps low-affinity residual interaction of mutant TRPV4 with RhoA.

However, in contrast to what was observed with WT TRPV4, expression of RhoA did not suppress hypotonic stress-induced calcium influx via neuropathy mutant TRPV4 (Fig. 4a–c), suggesting that the failure of mutant TRPV4 to interact with RhoA abrogates the inhibitory influence of RhoA on TRPV4 ion channel activity. Together, these data suggest that TRPV4–RhoA interactions modulate TRPV4 channel activity, and that this modulatory effect is disrupted by neuropathy-causing mutations.

**Binding of the intracellular N-terminal domain of TRPV4 to RhoA inhibits RhoA activation.** Given our finding that WT but not neuropathy mutant TRPV4 interacts with RhoA, we next sought to determine the functional consequences of this interaction for RhoA function. The finding that TRPV4 preferentially recognizes the GDP-bound, inactive conformation of RhoA suggested that TRPV4 might either be acting in a similar fashion to RhoA GEFs, which interact with GDP-bound RhoA to promote GTP exchange and RhoA activation, or alternatively, in a manner similar to the RhoA inhibitory protein RhoGDI, which functionally sequesters inactive RhoA and thereby prevents its activation[46]. To distinguish between these possibilities, we used the T-Rex-TRPV4 cells as they allow for controlled, uniform expression of both WT and mutant TRPV4. We used a well-described pull-down approach that utilizes a fragment of the RhoA-GTP binding protein Rhotekin conjugated to beads[47] to assess levels of active, GTP-bound RhoA in the presence or absence of TRPV4 expression. Cells were incubated with the specific TRPV4 antagonist HC067 to eliminate any potential confounding cytotoxic effects resulting from excessive influx of

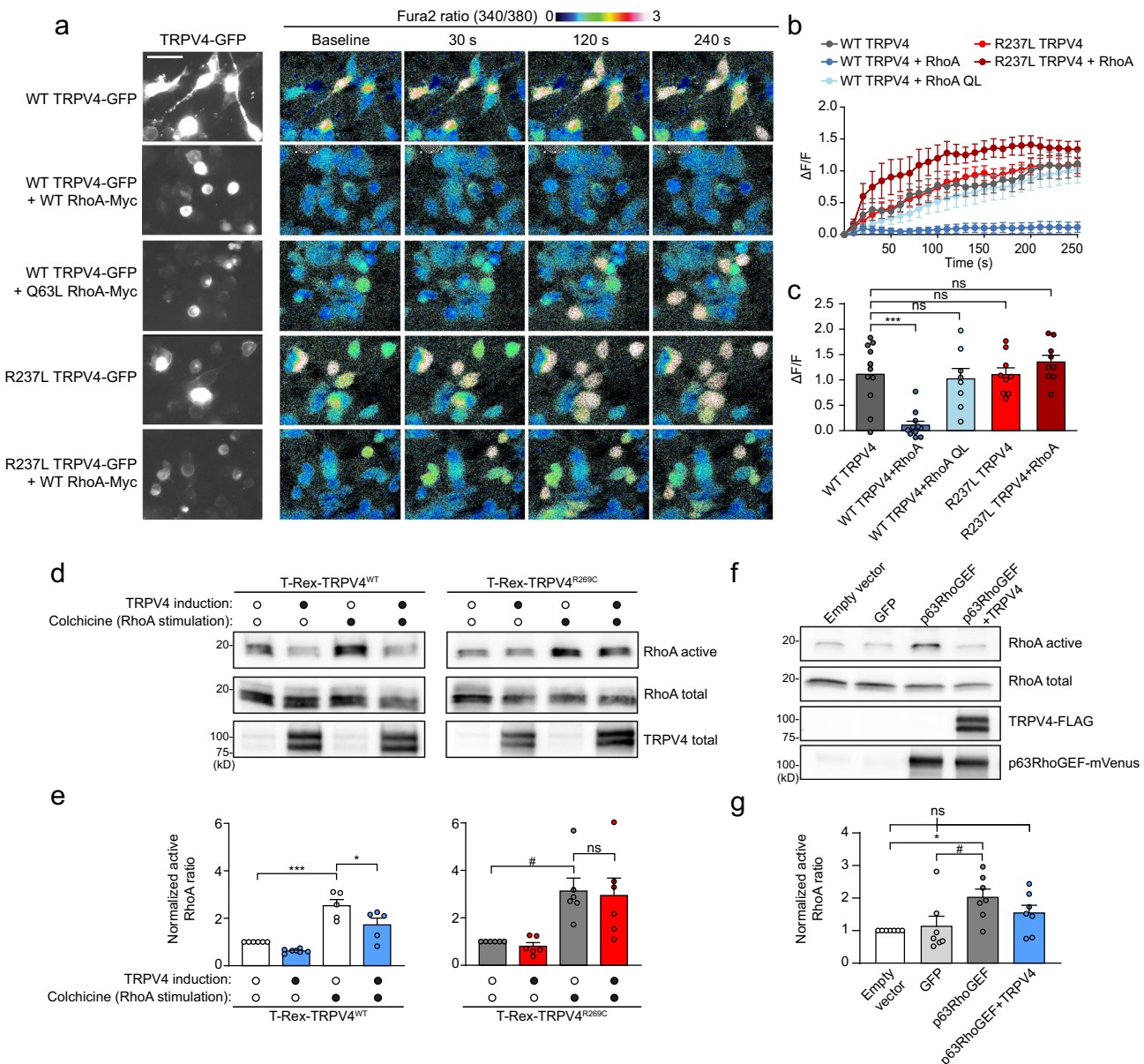

**Fig. 4 TRPV4–RhoA interactions result in reciprocal functional inhibition. a** MN-1 cells transfected with WT or R237L TRPV4-GFP alone or in combination with WT RhoA or constitutively active RhoA (Q63L) were loaded with calcium indicator Fura-2 AM and treated with hypotonic saline (30 mM NaCl) at time 0. WT RhoA inhibits TRPV4-mediated calcium influx of WT TRPV4 but not R237L TRPV4. RhoA-Q63L does not inhibit WT TRPV4-mediated calcium influx. Scale bar, 50 μm. **b** Change in Fura ratio (340/380) divided by baseline Fura ratio (ΔF/F) in experiments shown in **a**. $n = 9$ coverslips per condition. **c** Change in Fura ratio (340/380) divided by baseline Fura ratio at time = 250 s in experiments shown in **a**. One-way ANOVA with Tukey post hoc test, $n = 9$ coverslips per condition, ***$p = 0.0002$. **d** T-Rex-TRPV4 cells were uninduced or induced with tetracycline (15 ng/ml, 16 h), then treated with vehicle or RhoA stimulator colchicine (10 μg/ml, 45 min) followed by analysis of RhoA activation. WT TRPV4 expression reduces basal RhoA activation (lane 2 vs. 1) and colchicine-stimulates RhoA activation (lane 4 vs. 3), whereas neuropathy mutant TRPV4 fails to influence basal or stimulated RhoA activity. **e** Densitometry of band intensities (active RhoA/total RhoA); representative blots shown in **d**. Data for T-Rex-TRPV4^WT and T-Rex-TRPV4^R269C conditions were normalized to their respective uninduced and vehicle-treated condition from each cell line. WT TRPV4 but not neuropathy mutant TRPV4 inhibits RhoA activation. One-way ANOVA with Tukey post hoc test, $n = 6$, *$p = 0.046$, ***$p = 0.0004$, #$p = 0.030$. **f** HEK293T cells were transfected with empty vector, GFP, p63RhoGEF-mVenus, or p63RhoGEF-mVenus and WT TRPV4-FLAG in the presence of HC067 (0.5 μM), and RhoA activity was analyzed after 24 h. Expression of p63RhoGEF leads to increased RhoA activation that is inhibited by WT TRPV4. **g** Densitometry of active RhoA band intensity, normalized to empty vector transfection; representative blot shown in **f**. One-way ANOVA with Tukey post hoc test, $n = 7$, *$p = 0.028$, #$p = 0.020$. Data are presented as means ± SEM. Further details regarding the number of times experiments were repeated are presented in "Methods" under "Statistics and reproducibility".

calcium. We found that induction of WT TRPV4 expression in the presence of HC067 had an inhibitory effect on basal levels of active RhoA (Fig. 4d, e, compare lanes 1 and 2 of T-Rex-TRPV4^WT blot), suggesting that TRPV4 attenuates activation of RhoA. In contrast, neuropathy mutant TRPV4 did not inhibit

basal RhoA activation (Fig. 4d, e, compare lanes 1 and 2 of T-Rex-TRPV4^R269C blot), consistent with a requirement for binding of TRPV4 and RhoA for inhibition of RhoA activation.

Having observed basal inhibition of RhoA by TRPV4, we next investigated whether TRPV4 influences activation of RhoA in

response to stimulation. To test this possibility, we assessed active RhoA levels in T-Rex-TRPV4[WT] and T-Rex-TRPV4[R269] cells treated with the RhoA activator colchicine[46,47] and the TRPV4 antagonist HC067. Expression of WT TRPV4 attenuated colchicine-induced RhoA activation (Fig. 4d, e), whereas neuropathy mutant TRPV4 failed to exert an effect on RhoA activation in response to colchicine. Thus, our results suggest that TRPV4 functions similarly to RhoGDI to prevent activation of RhoA. We next reasoned that if TRPV4 were sequestering inactive RhoA, then RhoA activation in response to overexpression of a RhoA GEF should be inhibited by TRPV4. To test this hypothesis, we transfected HEK293T cells with p63RhoGEF[48,49] alone or in combination with TRPV4 in the presence of HC067. As expected, expression of p63RhoGEF led to an increase in the active GTP-bound fraction of RhoA (Fig. 4f, g), but co-expression of WT TRPV4 inhibited RhoA activation by p63RhoGEF (Fig. 4f, g). Together, these data suggest that RhoA binding to inactive TRPV4 results in RhoA sequestration and suppression of activation.

**TRPV4-mediated calcium influx activates RhoA.** As our results thus far demonstrated inhibition of RhoA when bound to the N terminus of inactive TRPV4, we next sought to address whether TRPV4 ion channel activity has an impact on RhoA activation. Prior work has shown that RhoA can be activated by conditions that increase intracellular calcium levels[50,51], suggesting that TRPV4 ion channel activity might lead to increased RhoA activation. To test whether TRPV4 channel activity could regulate RhoA activation, we first demonstrated that the selective TRPV4 agonist GSK101 caused increased intracellular calcium in T-Rex-TRPV4 cells (Supplementary Fig. 5a) as well as phosphorylation of ERK (Supplementary Fig. 5b) as has been previously described[52]. Treatment of these cells with GSK101 also led to a robust increase in the active fraction of RhoA (Fig. 5a, b), but no activation of Rac1 and a small but significant activation of Cdc42 (Supplementary Fig. 5c, e), consistent with a recent report[53]. To compare the spatial and temporal dynamics of RhoA activation in response to WT or mutant TRPV4 ion channel stimulation, we transfected T-Rex-TRPV4 cells with a FRET-based RhoA biosensor that displays increased FRET levels when RhoA is in the active, GTP-bound state[54]. Stimulation of TRPV4 with GSK101 led to a transient increase in RhoA FRET in cells expressing WT TRPV4, which was most pronounced near the cell perimeter and plasma membrane (Fig. 5c, d and Supplementary Movie 1). However, RhoA FRET was significantly greater with agonist stimulation of neuropathy mutant TRPV4, with more rapid onset, longer duration, and higher peak RhoA activation (Fig. 5c–e and Supplementary Movie 2). Increased RhoA FRET began within 30–60 s of TRPV4-mediated calcium influx (Fig. 5f, g and Supplementary Movie 3). With TRPV4 stimulation, we observed concomitant changes of the actin cytoskeleton with increased stress fiber formation, cellular process retraction, and cellular rounding (Fig. 5h, Supplementary Fig. 5f, and Supplementary Movies 4 and 5). We also observed reduced TRPV4–RhoA interaction by co-immunoprecipitation following activation of TRPV4 with hypotonic saline (Fig. 5i, j), but not by GSK101 (Supplementary Fig. 5g, h). Thus, our results are consistent with a model in which WT TRPV4 serves a dual function in which it can bind and inhibit RhoA when TRPV4 is in the inactive, closed state, but can also release and activate RhoA upon stimulation of TRPV4 ion channel activity. Furthermore, taken together with our findings described above, these data demonstrate that neuropathy mutations disrupt bidirectional functional interactions between TRPV4 and RhoA, with the net effect of increased RhoA

stimulation and loss of RhoA-dependent inhibition of mutant TRPV4 ion channel function.

**TRPV4 and RhoA interactions regulate cell morphology and are disrupted by neuropathy mutations.** As RhoA and its downstream effectors exert powerful control over cytoskeletal remodeling and neuronal morphogenesis[32,33,55], we sought to determine whether TRPV4 and its interactions with RhoA also affect cytoskeletal dynamics. In MN-1 cells, expression of TRPV4 alone promoted outgrowth of neurite-like processes, and this was markedly enhanced by inhibition of TRPV4 ion channel activity, consistent with a role for inactive TRPV4 in modulating cytoskeletal changes (Fig. 6a, b). TRPV4 strongly co-localized with actin, particularly within filopodia-like structures within neurites (Supplementary Fig. 6a). Neuropathy mutant TRPV4 failed to promote neurite growth, and this effect could not be rescued by ion channel inhibition with HC067 (Fig. 6a, b), suggesting an ion channel-independent disruption of cytoskeletal modulation in neuropathy mutants. In contrast, while expression of skeletal dysplasia mutations failed to promote neurite outgrowth in the absence of channel inhibition, neurite outgrowth could be completely rescued by blocking ion channel activity with HC067 (Fig. 6a, b). These results highlight a function of TRPV4 in regulating the cytoskeleton that occurs independent of ion channel activity, and suggest that neuropathy mutations specifically disrupt this function.

We next tested whether the observed TRPV4-dependent changes in the cytoskeleton were dependent on modulation of RhoA activity. Using the RhoA FRET biosensor to directly assess RhoA activation within MN-1 cells, we found that expression of WT and skeletal dysplasia mutant TRPV4, but not neuropathy mutant TRPV4, led to a significant decrease in RhoA FRET preferentially within the distal portion of neurite-like cellular processes (Fig. 6c, d and Supplementary Fig. 6b, c) as compared to cell bodies (Supplementary Fig. 6d–g). These results demonstrate a correlation between localized inhibition of RhoA activation and TRPV4-mediated augmentation of neurite outgrowth. Indeed, overexpression of RhoA strongly inhibited TRPV4-mediated neurite outgrowth (Fig. 6e, f). In addition, mild RhoA inhibition with a low dose of the potent and specific Rho inhibitor exoenzyme C3 transferase[56] potentiated the effect of WT TRPV4 on neurite outgrowth and also partially rescued the impaired neurite outgrowth with expression of neuropathy mutant TRPV4 (Fig. 6g, h), suggesting that the failure of neurite outgrowth with neuropathy mutations is at least in part due to excessive RhoA activity. Based on these collective results, we hypothesized that over-activation of RhoA might contribute to the pathogenesis of TRPV4 neuropathy, and that inhibition of RhoA could potentially rescue mutant TRPV4-mediated neuronal degeneration. To test this hypothesis, we used a fly model of TRPV4 neuropathy in which human TRPV4 is expressed under control of the UAS-Gal4 system[57]. This model demonstrates several mutation-dependent nervous system phenotypes, including failure of wing opening with pan-neuronal expression of mutant TRPV4 or limited expression within CCAP neurons whose activity is responsible for wing expansion; impaired locomotion with inducible pan-neuronal expression in adulthood; and disrupted axonal trafficking and degeneration of sensory neuron dendrites and axonal projections with expression within class IV dendritic arborization (C4da) neurons[26]. Notably, C4da neurons have also been used to model other forms of CMT in *Drosophila*[58]. Given the abnormalities of MN-1 neurite outgrowth with expression of neuropathy mutant TRPV4, we were particularly interested in examining the impact of RhoA on

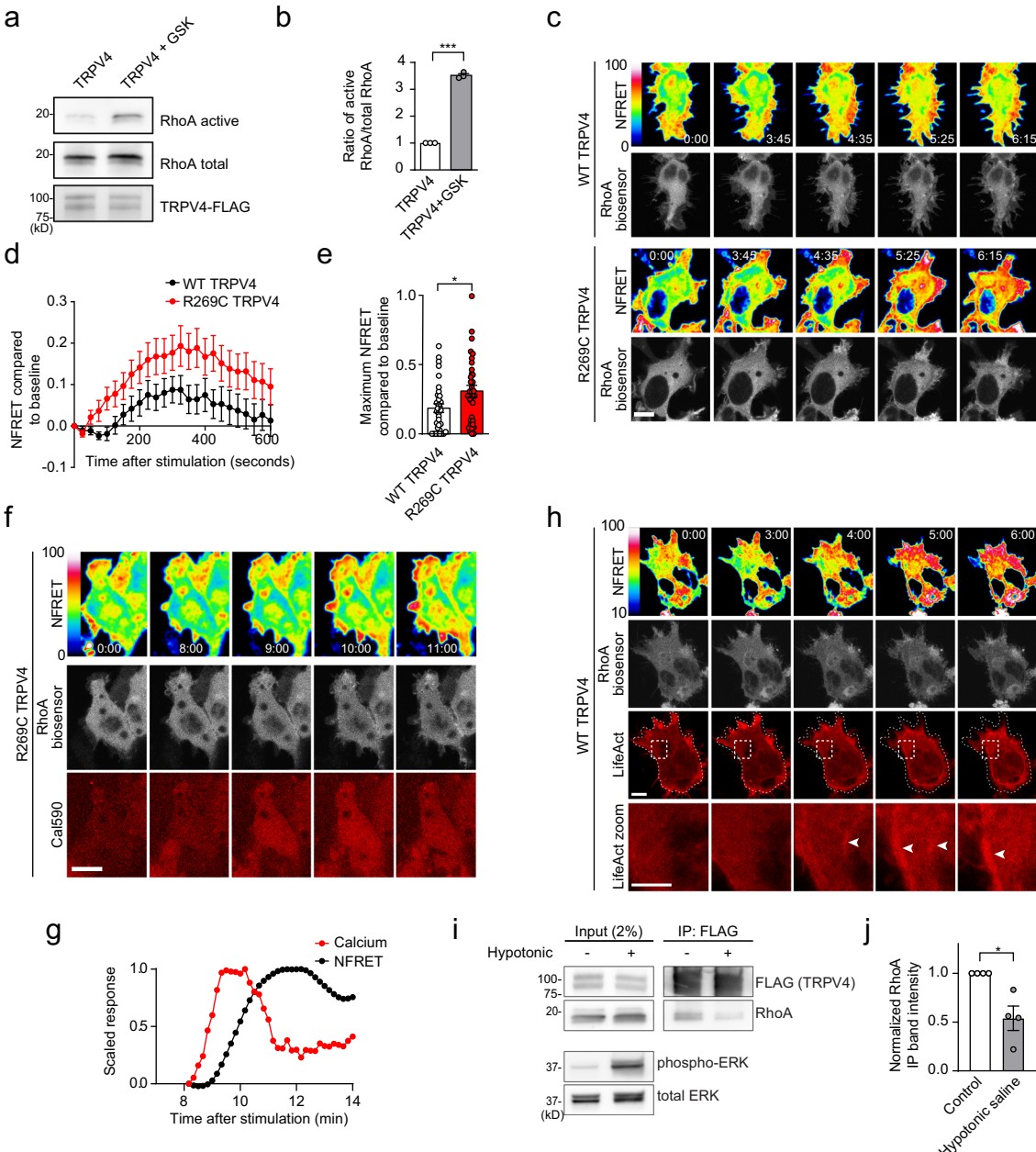

**Fig. 5 TRPV4-mediated calcium influx activates RhoA and cytoskeletal changes. a** RhoA activation assay of T-Rex-TRPV4 cells induced with tetracycline (15 ng/ml, 16 h) in the presence of HC067, followed by removal of HC067 and treatment with GSK101 (5 min). **b** Densitometry of band intensities (active RhoA/total RhoA) normalized to untreated condition; representative blot shown in **a**. Paired two-tailed $t$-test, $n = 3$, ***$p = 0.0007$. **c** T-Rex-TRPV4 cells were transfected with RhoA2G biosensor and induced with tetracycline (15 ng/ml, 16 h) in the presence of HC067. Upon HC067 removal, cells were treated with GSK101 and RhoA FRET was assessed for 10 min. Confocal images show normalized FRET (FRET signal/FRET donor intensity) (top) and RhoA biosensor distribution (mVenus, bottom). Scale bars, 10 μm. **d**, **e** Averaged NFRET time course (**d**) and average peak NFRET (**e**) compared to baseline of GSK101-stimulated RhoA NFRET in cells expressing WT TRPV4 or neuropathy mutant TRPV4; representative images in **c**. Unpaired two-tailed $t$-test, $n = 34$ cells, *$p = 0.0189$. **f**, **g** Time series of a R269C TRPV4-FLAG-expressing cell showing changes in calcium (Cal590) and NFRET following treatment with GSK101 at time 0 (**f**); intracellular calcium and NFRET signals shown over time (**g**). Scale bar, 10 μm. **h** Time series of a WT TRPV4-FLAG-expressing cell co-transfected with LifeAct-mCherry and treated with GSK101 at time 0 showing actin stress fiber formation (arrowheads) and cellular contraction. Scale bars, 10 and 5 μm for inset. **i** T-Rex-TRPV4^WT cells were induced in the presence of HC067 as in **a** followed by treatment with control media or hypotonic saline (5 min) followed by TRPV4 immunoprecipitation. Hypotonic saline causes partial disruption of TRPV4–RhoA interaction. Increased phospho-ERK (Thr202/Tyr204) indicates TRPV4-mediated calcium influx. **j** Densitometry of band intensities (immunoprecipitated RhoA/RhoA input); representative blot in **i**. Paired two-tailed $t$-test, $n = 4$, *$p = 0.035$. Data are presented as means ± SEM. Where indicated, HC067 was used at 0.5 μM; GSK101 at 100 nM. Further details regarding the number of times experiments were repeated are presented in "Methods" under "Statistics and reproducibility".

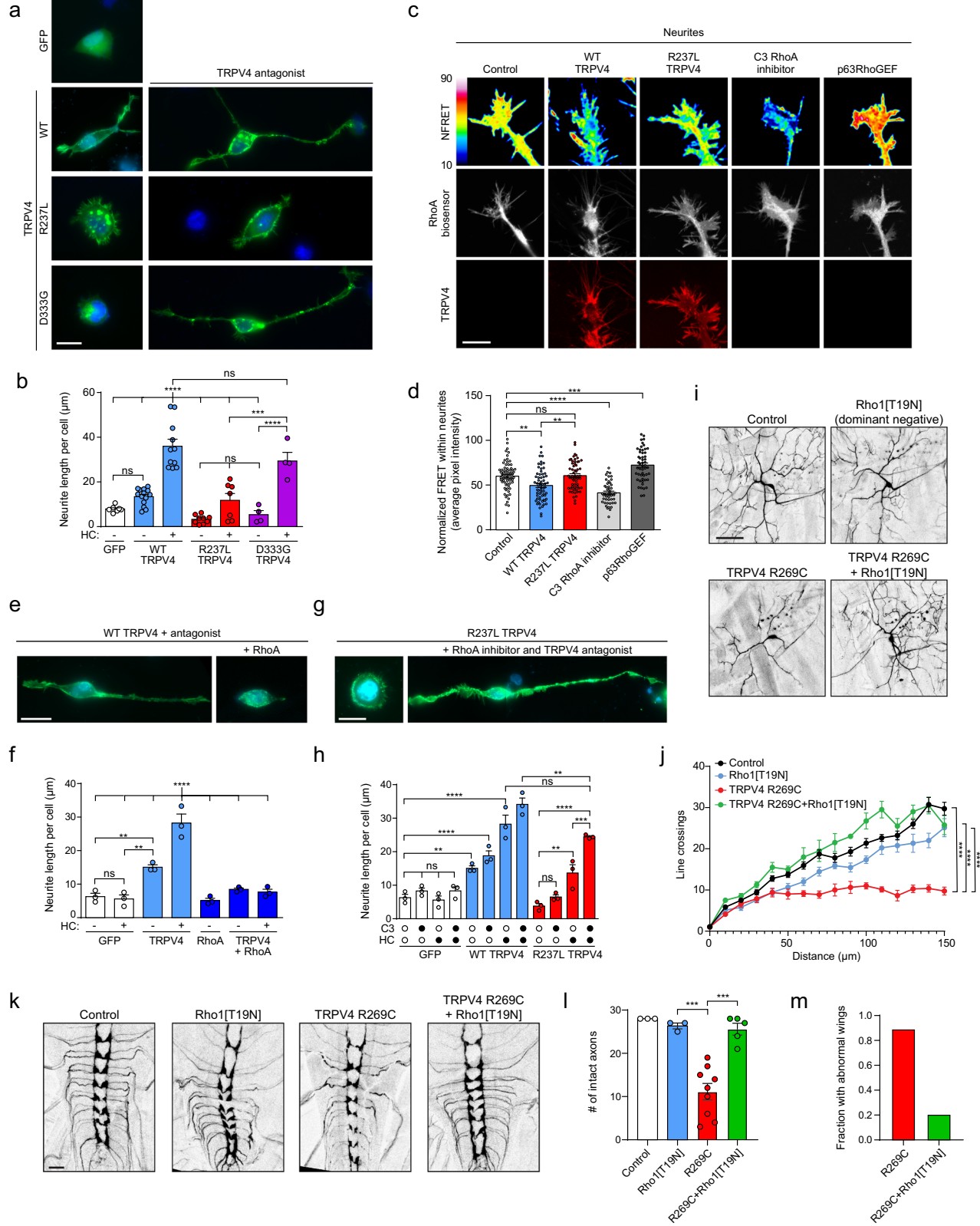

sensory neuron morphology and degeneration in the fly model. Strikingly, we found that genetic inhibition of the activity of the *Drosophila* RhoA ortholog Rho1 by expression of dominant negative Rho1[T19N] rescued both dendritic (Fig. 6i, j) and axonal (Fig. 6k, l) degeneration with expression of neuropathy mutant TRPV4. We also found strong rescue of wing opening failure with expression of Rho1[T19N] (Fig. 6m). Together, these

results suggest that neuropathy mutant TRPV4 can cause cytoskeletal disruption in vivo through excessive activation of RhoA, and these defects can be rescued by RhoA inhibition.

## Discussion

In this study, we demonstrate that mutations within the TRPV4 ARD specifically disrupt interaction with a crucial regulator of the

**Fig. 6 TRPV4 and RhoA interactions regulate cell morphology and are disrupted by neuropathy mutations.** Representative immunofluorescence images (**a**) and quantification of neurite length per cell (**b**) of MN-1 cells transfected with WT, R237L (neuropathy mutant), or D333G (skeletal dysplasia mutant) TRPV4-FLAG in the presence (right) or absence (left) of TRPV4 antagonist HC067. $n = 4$–14 independent experiments, ***$p = 0.007$, ****$p < 0.0001$. Scale bars, 20 μm. Representative images (**c**) and average NFRET (**d**) of neurites of MN-1 cells transfected with RhoA FRET biosensor. Cells were co-transfected with WT TRPV4- or R237L TRPV4-mScarlet in the presence of TRPV4 antagonist HC067 or p63RhoGEF-Myc or treated with C3 RhoA inhibitor (1 μg/ml, 2 h). Scale bars, 10 μm. $n = 47$–67 neurites, **$p = 0.0026$ (lane 1 vs. 2), **$p = 0.0020$, ***$p = 0.0005$, ****$p < 0.0001$. **e** Representative images of MN-1 cells transfected with WT TRPV4-FLAG with or without co-expression of WT RhoA-Myc in the presence of HC067. Scale bars, 20 μm. **f** Quantification of neurite length per cell with or without overexpression of RhoA and/or HC067 treatment. $n = 3$ independent experiments, **$p = 0.0039$ (lane 1 vs. 3), **$p = 0.0019$ (lane 1 vs. 4), ****$p < 0.0001$. **g** Representative images of MN-1 cells transfected with R237L TRPV4-FLAG in the presence of HC067 with (left) or without (right) low-dose C3 RhoA inhibitor (20 ng/ml). Scale bars, 20 μm. **h** Quantification of neurite length per cell in cells expressing WT or R237L TRPV4-FLAG with or without HC067 and/or low-dose C3 RhoA inhibitor (20 ng/ml). $n = 3$ independent experiments, **$p = 0.0086$ (lane 1 vs. 5), **$p = 0.0028$ (lane 8 vs. 12), **$p = 0.0022$ (lane 9 vs. 11), ***$p = 0.0007$, ****$p < 0.0001$. Confocal projections (**i**) and Sholl analysis (**j**) of C4da neuron dendrites from wandering third-instar larvae expressing UAS-CD8::GFP and TRPV4 R269C alone or in combination with dominant negative Rho1[T19N]. Scale bars, 50 μm. $n = 9$–17 neurons, ****$p < 0.0001$. Confocal projections (**k**) and quantification of intact axons in the first seven ventral nerve cord segments (**l**) of C4da axonal projections from wandering third-instar larvae expressing UAS-CD8::GFP and TRPV4 R269C alone or in combination with dominant negative Rho1[T19N]. Scale bars, 20 μm. $n = 3$–9 larvae, ***$p = 0.0004$ (lane 2 vs. 3), ***$p = 0.0001$ (lane 3 vs. 4). **m** Fraction of flies with abnormal unexpanded wings with expression of TRPV4 R269C alone or in combination with Rho1[T19N]. $n = 162$ (TRPV4 R269C) and 68 (TRPV4 R269C + Rho1[T19N]) flies. For **b**, **f**, **h**, **j**, **l**, statistics reflect one-way ANOVA with Tukey post hoc test. Data are presented as mean ± SEM. Where indicated, HC067 was used at 0.5 μM. Further details regarding the number of times experiments were repeated are presented in "Methods" under "Statistics and reproducibility".

actin cytoskeleton, RhoA, providing a potential link between the function of TRPV4 in cytoskeletal dynamics and the pathogenesis of human neurological disease. We demonstrate direct interaction of the TRPV4 ARD with inactive, GDP-bound RhoA, and further show how mutual TRPV4–RhoA inhibition and TRPV4 calcium-dependent activation of RhoA play important roles in modulating cellular outgrowth, both in cultured cells and in a fly model of TRPV4 neuropathy. Together, our results uncover a complex interplay between TRPV4 and RhoA in regulating cell morphology and suggest that increased RhoA activation may be an important pathologic consequence of TRPV4 neuropathy mutations.

**Multifaceted TRPV4–RhoA interactions regulate cytoskeletal dynamics**. The TRPV4 N-terminal ARD is predicted to serve as a platform for protein–protein interactions[12,17], but the only previously identified TRPV4 ARD interactors are calmodulin and ATP, whereas PACSIN isoforms bind directly to the N-terminal PRD that lies adjacent to the TRPV4 ARD[29,40,59]. Here, we identify RhoA as a direct TRPV4 ARD interactor and demonstrate that TRPV4–RhoA binding leads to reciprocal functional inhibition of both proteins. The mechanism for the inhibition of TRPV4 channel activity is unclear, but could involve modulation of TRPV4 N-terminal interactions with membrane phospholipids such as $PI(4,5)P_2$ or other proteins that control TRPV4 channel gating in a manner similar to PACSIN-mediated TRPV4 channel inhibition[15,40,45].

In addition to RhoA-mediated TRPV4 inhibition, we demonstrate a dual role for TRPV4 in regulating RhoA function, with TRPV4 binding-dependent inhibition as well as calcium-mediated activation of RhoA. We show that the TRPV4 ARD directly interacts with the effector binding switch loop regions of RhoA, leading to inhibition of RhoA activation by both chemical stimuli and GEF overexpression. Thus, binding by inactive TRPV4 functionally sequesters RhoA, analogous to the action of RhoGDI, thereby reducing the pool of RhoA available for activation[46]. On the other hand, stimulation of TRPV4 ion channel activity leads to both disruption of RhoA binding and subsequent RhoA activation that occurs rapidly in response to increased intracellular calcium. These results are consistent with prior work showing regulation of RhoA by calcium[50,51] and TRPV4 channel activation[60,61]. Calcium-mediated activation of RhoA is thought to involve phosphorylation of RhoA regulatory

proteins by calcium-sensitive kinases such as CaMKII[62–64]. This is particularly intriguing given our recent work showing that CaMKII inhibition is protective in *Drosophila* and cultured mammalian neurons expressing TRPV4 neuropathy mutants[26]. Together, these data suggest that CaMKII may be a critical downstream node that transduces TRPV4-mediated calcium influx to regulate RhoA activity and cytoskeletal changes.

RhoA function in cytoskeletal remodeling is subject to precise temporal and spatial regulation, and small changes in RhoA activity within discrete cellular microdomains, such as within axons or neuronal growth cones, can exert a large influence on localized cytoskeletal changes[31,33,55]. The ability of TRPV4 to both sequester inactive RhoA when the TRPV4 channel is silent, and also to release and activate RhoA through ion channel activity, provides a system in which TRPV4 serves a dual function to either promote cytoskeletal outgrowth through RhoA inhibition, or stimulate cytoskeletal process arrest/retraction through calcium-mediated RhoA activation. In this model, TRPV4, which is restricted in its tissue expression and generally less abundant than RhoA, could serve to concentrate inactive RhoA at the plasma membrane within specific subcellular regions and transduce local mechanical or other environmental stimuli to influence RhoA-dependent cytoskeletal changes. Consistent with such a spatially defined role for TRPV4–RhoA interactions, we demonstrate that TRPV4-mediated RhoA inhibition is particularly robust within highly dynamic neurite-like processes.

**TRPV4–RhoA interactions are specifically disrupted by neuropathy mutations**. We show that TRPV4–RhoA interactions are markedly disrupted by neuropathy-causing mutations within the TRPV4 N-terminal ARD, but not by skeletal dysplasia mutations on the opposing face of the ARD. In our NMR chemical shift perturbation assay, residue E54 in RhoA was particularly affected by WT TRPV4 ARD binding, but not by the presence of R269C TRPV4 ARD. This glutamic acid is unique to RhoA and is not present in the related Rho GTPases Rac1 and Cdc42, consistent with our results demonstrating specificity for interaction of TRPV4 with RhoA and not Rac1 or Cdc42. Strikingly, mutation of this glutamic acid residue abolished interaction with TRPV4, suggesting that E54 may be important for electrostatic interaction with positively charged arginine residues within the TRPV4 ARD that appear to be critical for RhoA binding. Our observations thus provide a potential structural explanation for the uniform

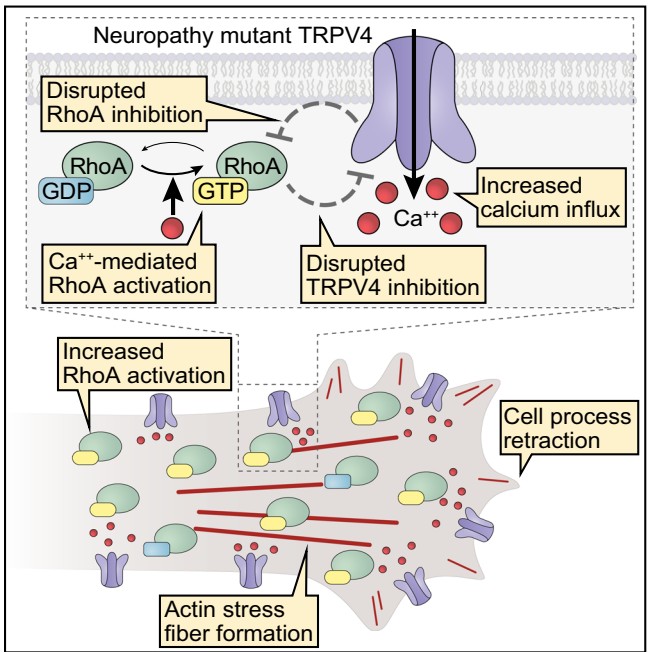

**Fig. 7 Neuropathy mutations cause multifaceted disruption of TRPV4–RhoA interactions.** Schematic representation of how TRPV4 neuropathy mutations disrupt normal TRPV4–RhoA functional interactions. TRPV4 binding to RhoA (Fig. 1) is disrupted by TRPV4 neuropathy mutations (Figs. 2 and 3). Impaired binding causes loss of TRPV4-mediated RhoA inhibition (Fig. 4) and disruption of RhoA-dependent TRPV4 ion channel inhibition (Fig. 4). Excessive calcium influx via neuropathy mutant TRPV4 causes further activation of RhoA (Fig. 5). Together, disrupted TRPV4–RhoA interactions lead to increased TRPV4 ion channel activity, increased RhoA activation, RhoA-mediated actomyosin contraction, actin stress fiber formation, and cell process retraction (Figs. 5 and 6).

disruption of RhoA interaction by neuropathogenic mutations within the TRPV4 ARD. Importantly, we also show that co-expression of neuropathy mutant TRPV4 with WT TRPV4 has an inhibitory effect on the interaction of WT TRPV4 with RhoA. These results suggest that TRPV4 neuropathy mutations can act in *trans* to disrupt TRPV4–RhoA interactions, even in the presence of a WT TRPV4 allele, consistent with the dominant mode of inheritance of TRPV4 neuropathy.

Our work elucidates several important consequences of the failure of TRPV4–RhoA interaction in neuropathy mutants (Fig. 7). First, mutant TRPV4 fails to provide inhibitory control of RhoA activation, thus allowing for an increase in the active fraction of RhoA. Second, stimulation of mutant TRPV4 leads to a more robust increase in the active fraction of RhoA as compared to WT TRPV4. Finally, the inhibitory effect of RhoA on WT TRPV4 ion channel activity is abolished by TRPV4 neuropathy-causing mutations. Taken together, our results demonstrate how failure of TRPV4–RhoA interactions due to neuropathy mutations serves to both directly and indirectly activate RhoA and exacerbate increased TRPV4 channel activity, thus creating a multi-hit scenario resulting in dysregulation of TRPV4-mediated calcium influx and RhoA-mediated cytoskeletal changes. Underscoring the pathological importance of increased RhoA activity, we find that TRPV4 neuropathy mutant-induced axonal and dendritic degeneration can be rescued in vivo by genetic inhibition of the *Drosophila* RhoA ortholog Rho1.

**Disrupted TRPV4–RhoA interaction and relevance to human disease**. A puzzling feature of TRPV4-related disease is the tissue specificity associated with human mutations despite similar gain of ion channel function with expression of both skeletal dysplasia and neuropathy mutants in heterologous systems. Our results suggest that the mechanism of gain of ion channel function with different classes of disease mutations may be distinct and therefore critical in understanding tissue-specific pathology. Specifically, disruption of protein–protein or protein–lipid interactions by neuropathy mutations may represent a fundamental pathological event that leads to tissue-specific alterations in ion channel gating and dysregulated cytoskeletal dynamics. Thus, uncoupling of neuropathy mutant TRPV4 from its normal binding partners and regulatory functions may contribute to the neuronal specificity in TRPV4-mediated neuropathy. In contrast, skeletal dysplasia mutations may cause disease through constitutive ion channel opening and stimulus-independent excess calcium influx[7,25], which are unlikely to be influenced by tissue-specific cytosolic interactions. Indeed, we show that interaction with RhoA and promotion of neurite outgrowth are preserved with skeletal dysplasia mutations.

Where might TRPV4–RhoA interactions potentially influence normal physiology and/or contribute to disease? Given that TRPV4 mutations cause axonal neuropathy, TRPV4–RhoA interactions may have a direct impact on neuronal development or maintenance, as suggested by our results in a fly model of TRPV4 neuropathy. In fact, RhoA, as well as its downstream effector ROCK, have been implicated as negative regulators of axonal outgrowth and regeneration[33,65–67]. Furthermore, the RhoA GEF proteins ARHGEF28, PLEKHG5, and ARHGEF10 each cause forms of Charcot–Marie–Tooth disease, underscoring that disruption of RhoA regulatory pathways is sufficient to cause peripheral nerve pathology[68–71]. Mutations in ARHGEF10 also cause inherited neuropathy in dogs with associated vocal fold weakness[72], a finding that is unusual in inherited neuropathy in general, but a defining feature of TRPV4-related neuropathy[2,73].

Alternatively, TRPV4 and RhoA interactions may be important in nonneuronal cells of the peripheral nervous system, including Schwann cells or vascular endothelial cells. Notably, some patients harboring ARHGEF10 mutations develop demyelinating neuropathy, suggesting that regulation of RhoA activity may be required for proper Schwann cell function[70]. TRPV4 function has perhaps been best characterized in vascular endothelial cells, where TRPV4 channel activation serves to regulate vascular smooth muscle tone, vasodilatory alterations in blood flow, and paracellular permeability through its influence on tight junction integrity[74,75]. Similarly, RhoA plays a critical role in vascular endothelial permeability through regulation of tight junction formation and maintenance[76,77]. In addition, recent work has suggested that TRPV4 and RhoA function in a common pathway to control endothelial barrier function both under normal conditions and in disease states associated with pathologic disruption of vascular integrity[78]. Our results suggest that targeting excessive TRPV4 and RhoA pathway activity may hold therapeutic promise for patients with TRPV4 mutations and in other conditions associated with TRPV4 and RhoA dysregulation, such as traumatic brain injury, stroke, and pulmonary edema[79–86]. Furthermore, with the availability of several highly specific TRPV4 antagonists, one of which has shown favorable tolerability in humans[87], pharmacologic TRPV4 inhibition may provide an opportunity to precisely limit pathological RhoA activation while preserving the wide range of vital RhoA functions in other tissues.

## Methods

**Antibodies and reagents**. Primary antibodies used were rabbit anti-FLAG (WB 1:1000, Cell Signaling Technology, 2368), mouse anti-FLAG (WB 1:1000, IF 1:1500, Cell Signaling Technology, 8146), rabbit anti-GFP (WB 1:1000, Cell Signaling Technology, 2555), rabbit anti-Myc (WB 1:1000, Cell Signaling Technology,

2272), mouse anti-Myc (WB 1:1000, IF 1:500, IP 5 μg/ml, Cell Signaling Technology, 2276), rabbit anti-RhoA (WB 1:1000, Cell Signaling Technology, 2117), mouse Rac1/2/3 (WB 1:1000, Cell Signaling Technology, 2465), and mouse Cdc42 (WB 1:1000, Cell Signaling Technology, 2462), rabbit anti-His (WB 1:1000, Cell Signaling Technology, 2365), rabbit anti-β-actin (WB 1:1000, Cell Signaling Technology, 4967), rabbit anti-RhoGDI (WB 1:1000, Cell Signaling Technology, 2564), rabbit phospho-Thr202/Tyr204 ERK1/2 (WB 1:1000, Cell Signaling Technology 9101), rabbit ERK1/2 (WB 1:1000, Cell Signaling Technology, 9102), mouse anti-FLAG M2 (IP 5 μg/ml, Sigma-Aldrich, F1804), rabbit anti-GFP (WB 1:1000, Thermo Fisher Scientific, A-11122), mouse anti-GFP (IP 5 μg/ml, Thermo Fisher Scientific, A-11120), rabbit TRPV4 (WB 1:1000, Cosmo Bio USA, KAL-KM119), rabbit anti-TRPV4 (WB 1:1000, IP 5 μg/ml, Abcam, ab39260), nonspecific rabbit IgG (IP 5 μg/ml, Cell Signaling Technology, 3900), and anti-mouse RhoA (WB 1:1000, Thermo Fisher Scientific, 1B3-4A10). Secondary antibodies used were HRP-conjugated monoclonal mouse anti-rabbit IgG, light chain specific (1:50,000, Jackson ImmunoResearch, 211-032-171), HRP-conjugated goat anti-mouse IgG, light chain specific (1:25,000, Jackson ImmunoResearch, 211-032-174), Alexa Fluor 488 goat anti-rabbit (1:1000, Thermo Fisher Scientific, A-11034), Alexa Fluor 488 goat anti-mouse (1:1000, Thermo Fisher Scientific, A-11029), Alexa Fluor 568 goat anti-rabbit (1:1000, Thermo Fisher Scientific, A-11011), and Alexa Fluor 555 goat anti-mouse (1:1000, Thermo Fisher Scientific, A-21422). Reagents used include C3 transferase (Cytoskeleton, Inc., CT03), HC067047 (Sigma-Aldrich, SML0143), GSK1016790A (Sigma-Aldrich, G0798), GDP (Sigma-Aldrich, 20-177), GTPγS (Sigma-Aldrich, 20-176). Uncropped images of all immunoblots can be viewed in the Source Data File.

**Mammalian expression plasmids**. TRPV4-FLAG in pcDNA3.1 was previously described[88]. TRPV4-GFP was generated by replacing the FLAG epitope tag with EGFP amplified from pEGFP-C1 (Clontech). TRPV4-M680K, TRPV4-R232C, TRPV4-R237L, TRPV4-R269C, TRPV4-I331F, TRPV4-D333G, TRPV4 P142A/P143L, TRPV4 ¹²¹AAWAA¹²⁵, RhoA E47A, and RhoA E54A with either FLAG or EGFP tags were generated by site directed mutagenesis using the Quikchange XL-II kit (Agilent, 200521) according to manufacturer protocols. TRPV4-mScarlet was generated from TRPV4-FLAG in pcDNA3.1 using Gibson assembly mix (New England Biolabs, E2611S) and PCR amplification of the mScarlet coding sequence from pmScarlet-C1 (Addgene, 85042). Primers used for mutagenesis and subcloning are listed in Supplementary Table 2. The following additional plasmids were obtained from Addgene: WT RhoA-GFP WT (12965), T19N RhoA-GFP (12967), Q63L RhoA-GFP (12968), WT RhoA-Myc (12962), T19N RhoA-Myc (12963), Q63L RhoA-Myc (12964), Cdc42-Myc (12972), RhoA2G-mTFP-mVenus (40176), mTFP-N1 (54521), mVenus-N1 (54640), and LifeAct-mCherry (67302). Expression plasmids for p63RhoGEF-mVenus and p63RhoGEF-Myc were a generous gift from Dr. Thomas Wieland (Universität Heidelberg). The expression plasmid for PACSIN1-Myc was a generous gift from Dr. Markus Plomann (University of Cologne).

**Liquid chromatography-mass spectrometry**. HEK293T cells were transfected with FLAG-tagged TRPV4 or empty vector with Lipofectamine LTX with Plus Reagent (Thermo Fisher Scientific) and lysed 24 h after transfection in IP Lysis Buffer (Pierce, 25 mM Tris-HCl pH 7.4, 150 mM NaCl, 1% NP-40, 1 mM EDTA, 5% glycerol) supplemented with Halt protease inhibitor cocktail (Thermo Fisher Scientific). Cells were lysed for 15 min followed by centrifugation at 17,000 × g for 10 min. Supernatants were precleared for 90 min at 4 °C in 50 μl of mouse IgG agarose beads (Sigma-Aldrich, A0919) followed by immunoprecipitation with EZview Red M2 anti-FLAG Affinity Gel (Sigma-Aldrich, F2426) for 2 h. Following several washes with IP wash buffer (PBS, 0.2% Tween 20), bound proteins were eluted with 300 μg/ml FLAG peptide (Sigma-Aldrich, F3290) in TBS for 30 min at 4 °C. Protein was then reduced with DTT and alkylated with Iodoacetamide, followed by precipitation with TCA/acetone and drying. Proteins were proteolyzed with trypsin (Promega) in 100 mM TEAB buffer at 37 °C overnight. Peptides were desalted on u-HJB Oasis plates (Waters), eluted with 60% acetonitrile/0.1% TFA, and dried. Digested peptides were analyzed by liquid chromatography interfaced with tandem mass spectrometry (LC-MS-MS) using Easy-LC 1000 (Thermo Fisher Scientific) HPLC system interfaced with a QExactive in FTFT. Peptides were loaded onto a C18 trap (S-10 μM, 120 Å, 75 μm × 2 cm, YMC, Japan) for 5 min at 5 ml/min in 2% acetonitrile/0.1% formic acid in-line with a 75 μm × 150 mm ProntoSIL-120-5-C18 H column (5 μm, 120 Å, Bischoff). Peptides eluting during the 2–90% acetonitrile in 0.1% formic acid gradient over 112 min at 300 nl/min were directly sprayed into a QExactive Plus mass spectrometer through a 1 μm emitter tip (New Objective) at 2.2 kV. Survey scans (full ms) were acquired from 350–1800 m/z with data dependent monitoring of up to 15 peptide masses (precursor ions), each individually isolated in a 1.6 Da window with −0.5 Da offset, and fragmented using HCD activation collision energy at 28 and 30 s dynamic exclusion. Precursor and fragment ions were analyzed at resolutions 70,000 and 35,000, respectively, and automatic gain control target values at $3 \times 10^6$ with 60 ms maximum injection time and $1 \times 10^5$ with 150 ms maximum injection time, respectively. Isotopically resolved masses in precursor (MS) and fragmentation (MS/MS) spectra were extracted from raw MS data without deconvolution and with deconvolution using Xtract or MS2 Processor in Proteome Discoverer software (Thermo Fisher Scientific, v1.4). All extracted data were searched using Mascot (Matrix Science, v2.5.1) against the RefSeq human protein database (release 72)

with the added enzymes and BSA, using the following criteria: sample's species; trypsin as the enzyme, allowing two missed cleavages; mass tolerance 8 ppm on precursor and 0.02 Da on fragment; cysteine carbamidomethylation, methionine oxidation, asparagine, and glutamine deamidation as variable modifications. Peptide identifications from Mascot searches were filtered at 1% false discovery rate confidence threshold, based on a concatenated decoy database search, using the Proteome Discoverer (Thermo Fisher Scientific). Prioritization of potential TRPV4 binding proteins was in part based on spectral counts as described in Supplementary Table 1. However, due to inherent limitations with quantification based on spectral counts, this approach has now largely been replaced by quantification based on precursor signal intensity, such as MS1 intensity measurements. The mass spectrometry proteomics data have been deposited to the ProteomeXchange Consortium via the PRIDE partner repository with the data set identifier PXD023758 and 10.6019/PXD023758[89,90].

**Co-immunoprecipitation**. MN-1 and HEK293T cells were cultured in Dulbecco's Modified Eagle's Medium (DMEM) supplemented with 10% (vol/vol) fetal calf serum (FCS) and penicillin/streptomycin at 37 °C with 6% CO₂. Cells were transfected with Lipofectamine LTX with Plus Reagent (Thermo Fisher Scientific) and lysed 24 h after transfection in IP Lysis Buffer (Pierce, 25 mM Tris-HCl pH 7.4, 150 mM NaCl, 1% NP-40, 1 mM EDTA, 5% glycerol, with or without 10 mM MgCl₂) supplemented with EDTA-free Halt protease inhibitor cocktail (Thermo Fisher Scientific). Cells were lysed for 15 min followed by centrifugation at 17,000 × g for 10 min. Supernatants were incubated with primary antibody bound to magnetic Protein G Dynabeads (Thermo Fisher Scientific) for 1 h at 4 °C followed by several washes in IP wash buffer (PBS, 0.2% Tween 20). To elute bound proteins, Laemmli sample buffer with β-mercaptoethanol was added to the beads and samples were heated for 10 min at 70 °C. Protein lysates were resolved on 4–15% TGX gels (Bio-Rad Laboratories) and transferred to PVDF membranes. Membranes were developed using SuperSignal West Femto Maximum Sensitivity Substrate (Thermo Fisher Scientific) and imaged using an ImageQuant LAS 4000 system (GE Healthcare). To reduce the appearance of IgG heavy chain and light in western blots, different antibody species were used for immunoprecipitation and western blotting whenever possible. In addition, the appearance of IgG heavy chain was reduced by using IgG light chain-specific HRP-conjugated secondary antibodies. Mouse choroid plexus was dissected from the lateral and third ventricles of freshly sacrificed C57BL/6J WT mice. Tissue was immediately placed in lysis buffer and processed as above. Supernatants were incubated with anti-TRPV4 polyclonal antibody (Abcam, ab39260) or control nonspecific rabbit IgG (Cell Signaling Technology, 3900) and incubated for 24 h at 4 °C followed by washing and elution as above. Mice were maintained according to protocols and ethical regulations approved by the Johns Hopkins University Institutional Animal Care and Use Committee. Conditions in the mouse facility were maintained as follows: humidity 42%, temperature 22.2 °C, and light–dark cycle 14:10 h.

**Co-incubation experiments**. To generate immunopurified TRPV4-FLAG and RhoA-GFP, HEK293T cells were transfected with polyethylenimine (PEI) (Sigma-Aldrich, 765090), and subject to co-immunoprecipitation as above, followed by five washes with IP wash buffer. TRPV4-FLAG was eluted from the Dynabeads by incubating the sample with 300 μg/ml FLAG peptide (Sigma-Aldrich, F3290) for 30 min at 4 °C on an orbital shaker. Immunopurified RhoA-GFP bound to Dynabeads was then co-incubated with immunopurified eluted TRPV4-FLAG for 1 h at 4 °C followed by three washes in IP wash buffer. Bound proteins were eluted and analyzed by western blot as above. For Coomassie staining, proteins were separated by SDS-PAGE followed by staining with SimplyBlue Safe Stain (Thermo Fisher Scientific, LC6060) according to manufacturer protocols.

**Immunofluorescence**. MN-1 and COS7 cells were grown on coverslips and transfected with Lipofectamine LTX for 18–24 h as above. Cells were then fixed for 15 min in PBS with 4% paraformaldehyde followed by three washes in PBS. Cells were permeabilized and blocked in blocking buffer (5% BSA and 0.3% TritonX-100 in PBS) for 30 min at room temperature. Cells were incubated with the primary antibody in blocking buffer overnight at 4 °C followed by incubation with secondary antibody in blocking buffer for 1 h at room temperature. Coverslips were mounted in Vectashield + DAPI (Vector Laboratories, H-1200) and imaged with a Zeiss 800 LSM confocal laser scanning microscope with a ×63 oil objective. Images were processed using Zen Blue 3.1 (Zeiss) and FIJI (NIH).

**Generation of stable inducible TRPV4-FLAG cell lines**. Stable, inducible TRPV4-FLAG 293 cells, referred to as T-Rex-TRPV4 cells, were generated using the Flp-In T-Rex system according to manufacturer protocols (Invitrogen). Briefly, WT TRPV4-FLAG and R269C TRPV4-FLAG in pcDNA3.1 were subcloned into pcDNA5/FRT/TO using the primers listed in Supplementary Table 2. Flp-In T-Rex 293 host cells were then co-transfected with TRPV4-FLAG pcDNA5/FRT/TO and pOG44 flp-recombinase. Stably integrated positive clones were selected with 100 μg/ml hygromycin (Invitrogen) and 15 μg/ml blasticidin (Invitrogen). Final polyclonal isogenic lines were tested for zeocin sensitivity (100 μg/ml) to confirm integration at the FRT site. Cells were cultured in standard media containing DMEM supplemented with 10% FCS. Every 2–3 passages, media was

supplemented with 15 μg/ml blasticidin to select against cells that had lost the transgenes. TRPV4-FLAG expression was induced by treatment with 15 ng/ml tetracycline (Invitrogen) for 16–20 h.

**Calcium imaging.** T-Rex-TRPV4 cells were induced with tetracycline (15 ng/ml) or MN-1 cells were co-transfected with pcDNA3-mCherry and pcDNA3.1-TRPV4-FLAG (WT or mutant) constructs using Lipofectamine LTX. Calcium imaging was performed on a Zeiss Axio Observer.Z1 inverted microscope equipped with a Lambda DG-4 (Sutter Instrument Company, Novato, CA) wavelength switcher. Cells were bath-loaded with Fura-2 AM (8 μM, Life Technologies) for 45–60 min at 37 °C in calcium imaging buffer (150 mM NaCl, 5 mM KCl, 1 mM MgCl₂, 2 mM CaCl₂, 10 mM glucose, 10 mM HEPES, pH 7.4). For hypotonic saline treatment, four volumes of NaCl-free calcium imaging buffer was added to one volume of standard calcium imaging buffer for a final NaCl concentration of 30 mM. For GSK101 treatment, GSK101 was added directly to the calcium imaging buffer to achieve the appropriate final concentration. Cells were imaged every 10 s for 30 s prior to stimulation with hypotonic saline or GSK101, and then imaged every 10 s for an additional 4 min. Calcium levels at each time point were computed by determining the ratio of Fura-2 AM emission at 340 nM divided by the emission at 380 nM. Data were expressed as either raw Fura ratio or change in Fura ratio from initial Fura ratio divided by initial Fura ratio (ΔF/F). Images were processed using AxioVision 4 (Zeiss) and FIJI (NIH).

**Cloning, expression, and purification of recombinant proteins**
*hV4-NTD.* The hV4-NTD (residues 2–397) was cloned from TRPV4 cDNA into a pET11a vector with an N-terminal His6-SUMO-tag via Gibson assembly. The construct was expressed in *E. coli* BL21-Gold(DE3) in TB medium supplemented with 0.04% (w/v) glucose and 100 μg/ml ampicillin. Cells were grown at 37 °C to an OD600 of 0.4, moved to 20 °C, grown to OD600 of 0.8 for induction with 75 μM IPTG and further grown at 20 °C for ~16 h. Cells were harvested via centrifugation and stored at −80 °C. For purification, cells were suspended in lysis buffer (20 mM Tris pH 8, 20 mM imidazole, 300 mM NaCl, 0.1% (v/v) TritonX-100, 1 mM DTT, 1 mM benzamidine, 1 mM PMSF, DNase, RNase, and SIGMAFast™ protease inhibitor cocktail) and sonicated on ice. The cell lysate was cleared via centrifugation and loaded onto a NiNTA gravity flow column (Qiagen). After washing (20 mM Tris pH 8, 20 mM imidazole, 300 mM NaCl), protein was eluted with 500 mM imidazole and dialyzed (20 mM Tris pH 8, 20 mM imidazole, 300 mM NaCl, 1 mM DTT, 1 mM EDTA) overnight at 4 °C in the presence of Ulp-1 protease (molar ratio 15:1). The tag-free hsV4-NTD protein was separated from the cleaved His6-SUMO-tag using a reverse Ni²⁺-NTA IMAC step and further purified by size exclusion chromatography (SEC) (HiLoad prep grade 16/60 Superdex200, GE Healthcare) in 20 mM Tris pH 7, 300 mM NaCl, 10% (v/v) glycerol, 1 mM DTT. All steps were carried out at 4 °C. After SEC, samples were flash frozen in liquid nitrogen and stored at −80 °C until further use. Protein purity was confirmed via SDS-PAGE and subsequent Coomassie staining.

*hV4-ARD-WT.* WT hV4-ARD (residues 148–397) was cloned from a pET21a vector containing hV4-NTD (residues 2–397) with a c-terminal His6-tag via Gibson assembly using primers listed in Supplementary Table 2.

*hV4-ARD-R269C.* hV4-ARD R269C was obtained by Quikchange-PCR using primers listed in Supplementary Table 2. For all PCRs, the KAPA HiFi PCR Kit (Kapa Biosystems) was used. The constructs were expressed in *E. coli* BL21-Gold(DE3) in LB medium and 100 μg/ml ampicillin. hV4-ARD-WT and hV4-ARD-R269C protein was produced as described above.

*RhoA.* A pET11a vector encoding human RhoA with an N-terminal His6-TEV-tag was obtained from GeneScript. The construct was expressed in *E. coli* BL21-Gold (DE3) in minimal medium supplemented with ¹⁵N-ammonium chloride (Cambridge Isotope Laboratories Inc.) as the sole nitrogen source for isotope labeling and 100 μg/ml ampicillin.
For purification, cells were suspended in lysis buffer (50 mM HEPES pH 7.5, 20 mM imidazole, 200 mM NaCl, 0,1% (v/v) TritonX-100, 10% glycerol (v/v), 1 mM benzamidine, 1 mM PMSF, DNase, RNase, and SIGMAFast protease inhibitor cocktail) and sonicated on ice. The cell lysate was cleared via centrifugation and the supernatant loaded onto a Ni²⁺-NTA gravity flow column (Qiagen). After washing (50 mM HEPES pH 7.5, 20 mM imidazole, 200 mM NaCl, 5% glycerol (v/v)), protein was eluted with high imidazole buffer (50 mM HEPES pH 7.5, 500 mM imidazole, 200 mM NaCl, 5% glycerol). The eluted protein was further purified by SEC (HiLoad prep grade 16/60 Superdex75, GE Healthcare) in 50 mM HEPES pH 7.5, 200 mM NaCl, 5 mM MgCl₂, 1 mM DTT, 1 mM EDTA, 1 mM PMSF. All steps were carried out at 4 °C. After SEC, samples were flash frozen in liquid nitrogen and stored at −80 °C until further use. Protein purity was confirmed via SDS-PAGE and subsequent Coomassie staining.

**GTPase activity assay.** Intrinsic GTPase activity of recombinant purified isotope-labeled ¹⁵N-RhoA was determined via the GTPase-Glo Assay (Promega, V7681) according to the manufacturer protocol. Luminescence was recorded with a FLUOStar Omega Microplate Reader (BMG Labtech). In this assay, GTPase activity is inversely correlated to the measured amount of luminescence.

**NMR spectroscopy.** NMR spectra of ¹⁵N-labeled RhoA (90 μM in SEC elution buffer supplemented with 10% D₂O) were recorded on a Bruker 600 MHz spectrometer equipped with a cryogenic triple probe at 25 °C. ¹H,¹⁵N 2D HSQCs of RhoA were processed using Bruker TopSpin 3.2 (Bruker, Karlsruhe). Comparison with previously published data allowed the transfer of 93.6% of backbone assignments and confirmed that RhoA was in the GDP-bound state[42]. For interaction studies with either hV4-ARD-WT or hV4-ARD-R269C, ¹⁵N-RhoA was mixed with the respective unlabeled proteins in a molar ratio of 2:1 (90 μM ARD:45 μM RhoA final concentrations). Integrals of respective spectra were obtained with Bruker TopSpin 3.2 and normalized on the ¹⁵N-RhoA spectrum without any hV4-ARD with the following equation:

$$\frac{I_{1,residue} - I_{0,residue}}{I_{0,residue}} = I_{rel,residue}$$

where $I_{0,residue}$ are the signal intensities of ¹⁵N-RhoA, $I_{1,residue}$ are the signal intensities of ¹⁵N-RhoA + hV4-ARD-WT, and $I_{rel,residue}$ are the resulting relative signal intensities changes, which were regarded as significant if the change exceeded twofold of the standard deviation (2σ).

**Circular dichroism spectroscopy.** CD spectra of all recombinant purified proteins (RhoA, hV4-NTD, hV4-ARD-WT, hV4-ARD-R269C) were measured on a Jasco-815 CD spectrometer (Jasco, Groß-Umstadt) with 1 mm quartz cuvettes at 25 °C in a spectral range between 195 and 260 nm with 1 nm scanning intervals, 1 nm bandwidth, and 50 nm/min scanning speed. Baseline corrections were performed automatically, and all spectra were obtained via automatic averaging of six measurements.

**Cell surface biotinylation.** HEK293T cells were cultured in 10-cm dishes, transfected with PEI, and processed for biotinylation assay 24 h after transfection. Cell surface biotinylation was performed with the Pierce Cell Surface Protein Isolation Kit (Thermo Fisher Scientific, 89881). Cells were briefly rinsed twice with PBS on ice before incubation with 0.25 mg/ml EZ-Link Sulfo-NHS-SS-Biotin in PBS with gentle rotation for 30 min at 4 °C. Cells were then rinsed three times with 50 mM glycine in PBS followed by lysis with RIPA buffer (Sigma-Aldrich) for 30 min at 4 °C. Lysates were sonicated at low power and incubated with NeutrAvidin Agarose slurry (Thermo Fisher Scientific, 29200) with end-over-end rotation for 2 h at 4 °C. The slurry was then washed on a spin column by centrifugation, and biotinylated proteins were eluted in Laemmli sample buffer supplemented with 50 mM DTT followed by western blot analysis.

**MN-1 neurite length quantification.** MN-1 cells were transfected, fixed, and stained as above. For quantification of neurite length, five randomly chosen fields per coverslip were imaged at ×20 magnification on an inverted Zeiss AxioVert 200 M fluorescence microscope. Images were then randomized and blinded, and neurites were traced manually using FIJI (NIH). For each image, the total length of neurites was divided by the number of transfected cells to generate neurite length per cell. The values of neurite length per cell from each image were then averaged and subjected to statistical analysis. For TRPV4 antagonist conditions, cells were incubated with 0.5 μM HC067 beginning at the time of transfection. For C3 transferase RhoA inhibitor conditions, cells were treated with 20 ng/ml C3 transferase for 5 h prior to fixation.

**RhoA, Rac1, and Cdc42 activation assays.** HEK293T or T-Rex-TRPV4 cells were grown in 10 cm tissue culture plates and either transfected or induced to express TRPV4 as above prior to processing with the RhoA Pull-Down Activation Assay kit (Cytoskeleton, Inc., BK-036), Rac1 Pull-Down Activation Assay kit (Cytoskeleton, Inc., BK-035), or Cdc42 Pull-Down Activation Assay kit (Cytoskeleton, Inc., BK-034) according to manufacturer protocols. In brief, cells were lysed in cell lysis buffer (50 mM Tris pH 7.5, 10 mM MgCl2, 0.5 M NaCl, 2% Igepal) followed by brief centrifugation and flash freezing in liquid nitrogen. Protein lysates were then thawed in a room temperature water bath, and 400 μg of lysate was incubated with 50 μg of pull-down beads for 1 h at 4 °C followed by a single wash with wash buffer. Bound proteins were eluted with Laemmli sample buffer and subjected to western blot analysis.

**RhoA FRET.** MN-1 cells or T-Rex-TRPV4 cells were grown in plates with coverslip bottoms (Mat-Tek) and transfected as above with RhoA2G-mTFP-mVenus. Cells expressing low levels of the RhoA biosensor were chosen for imaging analysis. FRET experiments were performed at the Johns Hopkins University SOM Microscope Facility, using a Zeiss LSM880 confocal with a Plan-Apochromat 63x/1.4 oil objective and incubation controls. Cells were incubated in calcium imaging buffer during imaging, and an argon-ion laser was used to excite RhoA2G-mTFP-mVenus at 458 or 514 nm, with detection bands at 455–515 nm and 515–580 nm. Images were then analyzed using the PixFRET plugin for FIJI[91]. In brief, bleed-through from mTFP and mVenus was determined using cells transfected with either fluorophore alone. For each image, background fluorescence from each channel was subtracted, and normalized FRET (NFRET) was determined by dividing raw FRET by the mTFP donor intensity. A threshold of 1.0 and Gaussian

blur of 2.0 were applied to the NFRET image. For MN-1 cell body and neurite FRET, a representative ROI was drawn in either compartment, and average NFRET within the ROI was determined using the Measure function in FIJI. For experiments monitoring RhoA FRET following TRPV4 agonist stimulation, T-Rex-TRPV4 cells were incubated in 0.5 μM HC067 followed by removal at the time of the experiment. Cells were then treated with 100 nM GSK101 and imaged every 5 s for 10 min. Baseline NFRET was subtracted from NFRET values at each time point and divided by baseline NFRET to generate values of NFRET compared to baseline (ΔNFRET/NFRET). For experiments monitoring actin dynamics following TRPV4 agonist stimulation, cells were co-transfected with LifeAct-mCherry followed by stimulation with 100 nM GSK101 as above. LifeAct-mCherry was visualized using an argon-ion laser to excite at 594 nm with detection band at 580–735 nm. For concurrent FRET and calcium imaging, cells were loaded with 2.5 μg/ml Cal590 (AAT Bioquest, 20511) for 1 h prior to imaging, then imaged as with LifeAct-mCherry.

**Drosophila sensory neuron imaging**. Flies were raised on standard cornmeal-molasses food. Experiments were performed at 25 °C with a 12/12 h day/night cycle. The following stocks were obtained from the Bloomington Stock Center: UAS-CD8::GFP, ppk-GAL4, and UAS-Rho1[T19N] (dominant negative). Transgenic UAS-TRPV4 R269C flies were previously described[26]. Larvae were filleted, processed, and imaged as previously described[26]. Wandering third-instar larvae were filleted to expose the CNS and body wall prior to fixation in 4% paraformaldehyde. Larvae were then immunostained with anti-GFP (Thermo Fisher Scientific, A-11120) followed by mounting onto glass slides in Fluoromount-G mounting media (Southern Biotech). Slides were imaged on a Zeiss 800 LSM confocal laser scanning microscope with a ×20 air objective. Image acquisition parameters were kept uniform across all samples in a given experiment. For dendrite morphology analysis, images were randomized and blinded, and all neurites were traced manually using FIJI (NIH). Images were then processed by Sholl analysis as previously described[26]. Branching was measured using the Sholl analysis tool in FIJI with Sholl radii of 10 μm. For quantification of axonal projections in the ventral nerve cord, images were randomized and blinded and the number of intact axons per segment in the first seven ventral nerve cord segments was counted (maximum score of 28).

**Drosophila wing expansion assay**. The wing expansion assay was carried out as previously described[26]. Flies carrying UAS-TRPV4 R269C or Rho1[T19N] were combined with w[1118];CCAP-GAL4/TM6B flies. Progeny were scored by eye using a stereomicroscope. Fly wings were scored as fully expanded, partially expanded, or fully unexpanded. For analysis, flies with partially expanded wings were counted as 0.5 normal and 0.5 unexpanded. Flies that had phenotypic traits of virgins at the time of initial collection were set aside and scored at least 4 h later.

**Statistics and reproducibility**. Statistical analysis was performed using Prism (Graphpad, v8). Details of statistical methods and tests employed are provided in the figure legends. Further details regarding the number of times individual experiments were repeated as well as the nature of $n$ reported in the figures are as follows: Fig. 1 representative blot from >10 (a) and three (b) independent experiments, (c) representative blot from at least four independent experiments, (d) representative image from three independent experiments, (e) representative blot from three independent experiments, (g) representative blot from two independent experiments, (h) representative image from three independent experiments, (j) $n =$ 4 independent experiments. Figure 2f representative blot from three independent experiments, (g) $n = 4$ (lane 5), 5 (lane 4), or 7 (lanes 1, 2, and 5) independent experiments. Figure 3a representative blot from three independent experiments, (b) representative blot from three independent experiments, (f) representative blot from three independent experiments. Figure 4b graphed values represent the average Fura ratios over time from $n = 9$ independent coverslips per condition. All cells in the single imaged field from each coverslip (between 20 and 40 cells) were averaged together to give a single data point for each coverslip, (c) $n = 9$ independent coverslips per condition as described above, (e) $n = 6$ independent experiments, (g) $n = 7$ independent experiments. Figure 5b $n = 3$ independent experiments, (d) $n = 34$ cells from three independent experiments, (e) $n = 34$ cells from three independent experiments, (j) $n = 4$ independent experiments. Figure 6b from left to right, $n = 7, 14, 11, 9, 7, 4,$ and 4 independent coverslips with quantification performed from five blindly chosen fields per coverslip, (d) $n = 47–67$ neurites per condition, with one neurite chosen per cell body, from four independent experiments (control $n = 67$, WT TRPV4 $n = 63$, R237L $n = 58$, C3 $n = 47$, p63RhoGEF $n = 52$), (f) $n = 3$ independent coverslips per condition with quantification performed from five blindly chosen fields per coverslip, (h) $n = 3$ independent coverslips per condition with quantification performed from five blindly chosen fields per coverslip, (j) from top to bottom, $n = 17, 9, 12,$ and 10 neurons from between five and nine larvae per genotype, (l) from left to right, $n = 3, 3, 9,$ and 5 larvae per genotype, (m) $n = 162$ flies for TRPV4 R269C, 68 flies for TRPV4 R269C + Rho1[T19N].

**Reporting summary**. Further information on research design is available in the Nature Research Reporting Summary linked to this article.

## Data availability

All data supporting the findings of this study and unique biological materials used in this study are available from the corresponding authors upon reasonable request. The mass spectrometry proteomics data have been deposited to the ProteomeXchange Consortium via the PRIDE partner repository with the data set identifier PXD023758[89,90]. Source data are provided with this paper.

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

## Acknowledgements

This work was supported by K08 NS102509 (B.A.M.), Inherited Neuropathies Consortium Fellowship (B.A.M.), American Academy of Neurology Neuroscience Research Training Fellowship (B.A.M.), TransMED Ph.D. Fellowship (E.D.), Fulbright Ph.D. Fellowship (E.D.), Sibylle Kalkhof-Rose Stiftung Fellowship (E.D.), NIH (NINDS) F31 NS105404 (W.H.A.), Max Planck Graduate Centre (MPGC) Ph.D. Fellowship (B.G.), Centre for Biomolecular Magnetic Resonance (BMRZ), Goethe University Frankfurt, funded by the state of Hesse (U.A.H.), the Naturwissenschaftliches-Medizinisches Forschungszentrum (NMFZ) Mainz (U.A.H.), NIH (NINDS) R01 NS094239 and NS082563 (T.E.L.), NIH (NINDS) R01 NS115475 (C.J.S.), and Muscular Dystrophy Association 629305 (C.J.S). Stocks obtained from the Bloomington *Drosophila* Stock Center (NIH P40ODO18537) were used in this study. We thank the Johns Hopkins Institute for Basic Biomedical Sciences Microscope Facility and Hoku West-Foyle for assistance with FRET image acquisition and analysis. This facility was supported by the Office of the Director and the NIGMS of the NIH under award number S10OD023548. We thank the Johns Hopkins Mass Spectrometry and Proteomics Facility for assistance with liquid chromatography-mass spectrometry data acquisition and analysis. We thank Natalia Nedelsky for paper editing and graphical assistance.

## Author contributions

Conceptualization: B.A.M., E.D., U.A.H., T.E.L., and C.J.S.; formal analysis: B.A.M. and E.D.; investigation: B.A.M., E.D., J.M.S., W.A.A., D.J.R., and A.R.L.; methodology: J.M.S., W.A.A., N.W.Z., and B.G.; resources: T.E.L.; writing—original draft: B.A.M.; writing—review and editing: B.A.M., E.D., T.E.L., and C.J.S.; visualization: B.A.M.; supervision: B.A.M., U.A.H., T.E.L., and C.J.S.; funding acquisition: B.A.M., U.A.H., T.E.L., and C.J.S.

## Competing interests

The authors declare no competing interests.
