## [Peer Review File · Nature Communications]

Reviewers' Comments:

Reviewer #1:

Remarks to the Author:

The work by McCray and cols. addresses the interaction between RhoA and TRPV4 in an attempt to explain the pathophysiology of disease causing TRPV4 mutations.

The authors combined imaging with classical biochemical techniques and state-of-the-art NMR to determine the characterize the interaction between TRPV4 and RhoA. Then they analyse the two-directional effect of such interaction, to show mutual inhibitory effects. This finding is particularly appealing to me as more long ago we obtained similar results (RhoA-TRPV4 interaction that resulted in lower RhoA activity) but as it went against the accepted view at that time (TRPV4 activates RhoA), we did not continue further. Authors finally analysed these results in the context of neuropathy causing TRPV4 mutations.

This is a really interesting manuscript that assembles a variety of novel data to elucidate molecular links between cytoskeleton controlling Rho GTPase and TRPV4. These data should also help to explain effects of TRPV4 mutations in human channelopathies. I recommend publication once several points have been addressed.

Pag 8 first parag. Related to figure 2. It says that HC did not modify RhoA-TRPV4 interaction but no data shown.

I331F seems to also affect V4-RhoA interaction (at least the WT and a trend with the M680K mutant), any explanation?

Does the activation of TRPV4 implies separation of TRPV4 from RhoA in order to promote RhoA activation?

Is it possible that RhoA also uses other regions within the N-term, different to the ARD, to interact with and modulate RhoA? Our own unpublished data suggest that altering the PIP2-binding site on the N terminus abolished the inhibitory effect of TRPV4 upon RhoA.

The effect of TRPV4 on neurite outgrowth is puzzling. While previous studies reported that activation of TRPV4 promoted neurite outgrowth (Jang...Oh, JBC 2012) while other show retraction (Goswami PlosOne 2010), now McCray and cols. claim a negative effect of TRPV4. Also in relation to these experiments it may be better to use the Sholl index to quantify neurite outgrowth.

Miguel A. Valverde

Reviewer #2:

Remarks to the Author:

Comments for McCray et al

This paper describes the novel interaction between TRPV4 channel and small GTPase RhoA. The authors found the interaction through the immunoprecipitation combined with MASS spec, and further identified the interaction by NMR. Interestingly, the disease related mutation of TRPV4 reduced the interaction with RhoA. The interaction appears to be important for the neurite extension. These findings are interesting; however, it is not clear whether the interaction really happens in cells for the proposed function. I would like to propose some approaches to strengthen the authors findings.

1. Both RhoA and TRPV4 are important for neuronal function, the synergy by the interaction should be examined. The authors performed NMR analysis, and thus there should be a lot of candidate residues to be mutated. Among them, the residues that are not reported for the dysfunction of TRPV4 might be included. Then it is possible to demonstrate that only the RhoA binding is changed by the mutation through the immunoprecipitation analysis combined with the Western blots or MASS spec, as authors performed for the identification of the RhoA as TRPV4 binding protein. These mutations in TRPV4 would further strengthen the importance of the interaction.
2. Similar approach can be possible from RhoA. The RhoA mutations that might affect (only) the

TRPV4 binding with lesser effect to the other binding proteins would be possible. But it would be difficult approach as compared to that from TRPV4 because of the abundance of RhoA binding proteins.

3. The binding of Cdc42 was examined and shown to be negative. The binding of TRPV4 to Rac should also be examined.

4. It is interesting that the TRPV4 binds to the GDP-bound form of RhoA. However, majority of GDP-bound RhoA should be in the cytoplasm. The co-localization of dominant-negative RhoA with TRPV4 should be shown. If it is not clear, then some complementary approaches, such as fluorescent lifetime and proximity ligation, and so on would be possible.

5. If TRPV4 inhibits RhoA upon TRPV4 activation, then it might be interesting mechanism for delaying the RhoA activation among Rho family GTPases. Then the time course of the activation of RhoA upon TRPV4 activation should be compared with the other small GTPases such as Cdc42 and Rac. RhoA might be simultaneously activated with Cdc42 and Rac in the absence of TRPV4 function.

Reviewer #3:

Remarks to the Author:

The manuscript by McCray et al. investigates in a very detailed way the interactions between TRPV4 and RhoA. TRPV4 interacts with the inactive, GDP-bound RhoA and the interaction is situated in the ankyrin repeat domain of TRPV4. This interaction is specifically disturbed by the presence of mutations in TRPV4 that induce neuropathies. RhoA binding to TRPV4 inhibits calcium influx, while influx of calcium activates RhoA. The model proposed by the authors is that TRPV4 serves a dual role. It can bind and inhibit RhoA when TRPV4 is in the closed state, while RhoA is activated once calcium is entering the cell upon stimulation of the TRPV4 channel. This bidirectional functional interaction is disturbed by neuropathy-inducing mutations. Last but not least, it is shown that the TRPV4-RhoA interaction regulates cell morphology which is disrupted by the neuropathy mutations. Inhibition of RhoA can restore neurite length in cells and in *Drosophila* larvae and RhoA inhibition is proposed as a potential therapeutic strategy. Overall, this manuscript contains an overwhelming amount of novel data and convincingly links RhoA to TRPV4 and provides a potential explanation for the effect of neuropathy-inducing mutations in TRPV4.

Detailed information is provided on every aspect of this study and all statistical assays are correct. This manuscript constitutes a very relevant addition to what is currently known about the potential mechanism behind the TRPV4 mutations causing peripheral neuropathies. I have a number of remarks but none of them questions the relevance of this work.

Major remarks

- The rationale for selecting RhoA from the mass spec data is not completely clear. There are many more hits present and in some cases the difference between WT and mutant is higher than for RhoA. Moreover, the published yeast-two hybrid screen also detects many more interactors. Not so clear why exactly RhoA was selected and the question arises whether these other hits could also play a role in the disease process.

- The impression exists (and it is also mentioned in the discussion) that there is a much closer connection between the observations recently published by the same research group in *Nature Communications* and the ones reported in this manuscript. The link between CamKII and RhoA could eventually be investigated in more detail using the fly models that are available (which could be a follow-up study).

- The interaction between RhoA and TRPV4 is always shown in overexpression systems (or in cell free systems). A crucial question is whether similar interactions also occur in cells when both TRPV4 and RhoA are expressed at physiological levels. If this is technically impossible to investigate, this important limitation should be discussed in more detail.

- The fly data are limited to the investigation of the impact of a dominant negative overexpression of Rho1 on the morphology of sensory neurons in the fly larvae. It would be interesting to know

whether this also affects the other phenotypes observed in the mutant TRPV4 flies.

Minor remarks

- If possible, MW markers should be indicated next to the Western blots.
- Fig. 1a,b What is the explanation for the fact that the input is so different for the experimental condition that is similar on both blots? It is also remarkable that the intensity of the GFP band is so low in panel a, while the FLAG band in panel b is so intense. This doesn't seem to correlate with the input.
- In COS7 cells (Suppl. Fig. 1g) RhoA seems to be enriched in the nucleus (or at least a perinuclear region). Which is the explanation for this?
- Fig. 1g: The effect of EDTA is difficult to judge as the condition without EDTA is missing. In the absence of this control, it is also difficult to judge whether the EDTA + excess Mg²⁺ condition leads to more binding.
- Only the intensity of the blot shown in Fig. 1j is quantified. Not clear why that is not systematically done with all the IP blots on this Figure. The number of blots quantified should also be indicated in the legend.
- Not so clear whether the results using the TRPV4 antagonist are indeed shown on Fig. 2.
- Text on Fig. 3e is difficult to read.
- The transfection efficiency of TRPV4 in the calcium experiment seems to be rather low. Moreover, the impression exists that also calcium increases are observed in cells that are TRPV4 negative. Is this a technical issue?
- If possible, the supplementary Figure 7 should be integrated in the manuscript as it is very helpful to understand the conclusions of this paper. In line with my major remark and if possible, CamKII should also be integrated in this figure.
- The discussion is a bit too long and should focus more on the most important aspects of this study
- Scale bars are missing on the pictures present in the supplementary material.
- Ref 26 is incomplete.

We appreciate the reviewers' positive comments and suggestions for improvement. In response, we have extensively revised the manuscript with the inclusion of new data and text modifications as outlined below in a point by point response to the reviewers' comments. We have added several panels of new data (**Fig. 3f, Fig. 5i-j, Fig. 6k-m, Supplementary Fig. 1e-h, Supplementary Fig. 3f, and Supplementary Fig. 5c-e**) including co-immunoprecipitation of endogenous TRPV4 and RhoA (**Supplementary Fig. 1f**). **Where new data figures have been added, the figure number and letter is marked in bold text.** Together, we believe these changes have significantly improved the manuscript, and we look forward to the reviewers' responses to the revised manuscript.

Reviewer #1:

The work by McCray and cols. addresses the interaction between RHOA and TRPV4 in an attempt to explain the pathophysiology of disease causing TRPV4 mutations.

The authors combined imaging with classical biochemical techniques and state-of-the-art NMR to determine the characterize the interaction between TRPV4 and RhoA. Then they analyse the two-directional effect of such interaction, to show mutual inhibitory effects. This finding is particularly appealing to me as more long ago we obtained similar results (RhoA-TRPV4 interaction that resulted in lower RhoA activity) but as it went against the accepted view at that time (TRPV4 activates RhoA), we did not continue further. Authors finally analysed these results in the context of neuropathy causing TRPV4 mutations.

This is a really interesting manuscript that assembles a variety of novel data to elucidate molecular links between cytoskeleton controlling Rho GTPase and TRPV4. These data should also help to explain effects of TRPV4 mutations in human channelopathies. I recommend publication once several points have been addressed.

We appreciate the reviewer's very positive comments and recommendation that this work be published. We have addressed the specific reviewer comments and suggestions below.

1. Page 8 first parag. Related to figure 2. It says that HC did not modify RhoA-TRPV4 interaction but no data shown.

We thank the reviewer for pointing out a lack of clarity in how the data was originally described, and we have now modified the text accordingly. The co-immunoprecipitation blot showing the interaction of RhoA and TRPV4 in the absence of antagonist is presented in the supplemental data (Supplementary Fig. 2a), while co-immunoprecipitation blots and quantification in the presence of antagonist or the ion channel pore inactivating mutation (M680K) are presented in Figure 2b-e. The text describing these data has now been modified to read as follows: "*we performed TRPV4-RhoA co-immunoprecipitation experiments in the presence (Fig. 2b-e) or absence (Supplementary Fig. 2a) of TRPV4 ion channel inhibition.*"

2. I331F seems to also affect V4-RhoA interaction (at least the WT and a trend with the M680K mutant), any explanation?

In our experimental systems, we have consistently observed that expression of the I331F mutant TRPV4 results in the highest level of cellular toxicity and most abnormal subcellular localization compared to other skeletal dysplasia mutants. This cellular toxicity may cause indirect disruption of TRPV4-RhoA interactions. Consistent with this interpretation, the interactions of I331F and RhoA are enhanced when the pore inactivating M680K mutation is introduced into TRPV4 (see Fig. 2d vs. 2e). In contrast, none of the neuropathy mutant forms of TRPV4 show an increased interaction with RhoA with introduction of the M680K mutation (see Fig. 2d vs. 2e).

3. Does the activation of TRPV4 imply separation of TRPV4 from RhoA in order to promote RhoA activation?

We appreciate this very insightful comment. As the reviewer suggests, our working model is that TRPV4 binds and inhibits RhoA, but that TRPV4 can activate RhoA when TRPV4 ion channel activity is stimulated. We have also theorized that release or “separation” of RhoA could occur in response to a specific TRPV4 stimulus. To investigate this possibility, we used T-Rex TRPV4^{WT} cells to assess co-immunoprecipitation of TRPV4-FLAG and endogenous RhoA, either under control conditions or after TRPV4 channel activation with GSK101 or hypotonic saline. Remarkably, we found that the mechanical stimulus of hypotonic saline led to disruption of RhoA binding (**Figure 5i-j**), whereas the synthetic agonist GSK101 did not (**Supplementary Fig. 5g-h**). For both stimulation paradigms, we confirmed TRPV4 channel activation by demonstrating increased phospho-ERK signal in the stimulated conditions (**Fig. 5i** and **Supplementary Fig. 5b**). Thus, we now show that physical separation of TRPV4 and RhoA can occur in response to hypotonic stress. We have now incorporated these findings in the results section as follows: “We also observed reduced TRPV4-RhoA interaction by co-immunoprecipitation following activation of TRPV4 with hypotonic saline (**Figure 5i-j**), but not by GSK101 (**Supplementary Fig. 5g-h**). Thus, our results are consistent with a model in which WT TRPV4 serves a dual function in which it can bind and inhibit RhoA when TRPV4 is in the inactive, closed state, but can also release and activate RhoA upon stimulation of TRPV4 ion channel activity.”

4. Is it possible that RhoA also uses other regions within the N-term, different to the ARD, to interact with and modulate RhoA? Our own unpublished data suggest that altering the PIP₂-binding site on the N terminus abolished the inhibitory effect of TRPV4 upon RhoA.

The reviewer makes an excellent point that while we show that the TRPV4 ARD is sufficient for RhoA interaction, there may in fact be additional regions/domains of TRPV4 that can influence RhoA binding. We have now investigated whether other regions of the TRPV4 N terminus affect co-immunoprecipitation of TRPV4 and RhoA. As prior work has demonstrated the importance of both the PIP₂ binding site and the proline rich domain (PRD) for TRPV4 ion channel function and protein-protein interactions¹⁻³, we tested whether these N-terminal domains influence RhoA interaction. In our hands, neither mutation of the PIP₂ binding site nor key proline residues in the PRD (which disrupt PACSIN interaction) had an impact on the binding of TRPV4 and RhoA (**Supplementary Fig. 3f**). While these experiments do not show a clear role for these domains in regulating RhoA interaction, they do not fully exclude the possibility that these and/or other domains may be involved in RhoA binding. These experiments are now described in the results as follows: “We also interrogated whether domains in the TRPV4 N terminus outside of the ARD

could influence RhoA interaction. We examined a PI(4,5)P₂ binding domain mutant (¹⁰⁷AAWAA¹¹¹) that alters channel activity and function^{1,2} as well as a PRD mutant (P142A/P143L), which has been shown to interfere with PACSIN interaction³. Neither of these mutants caused significant disruption of RhoA binding (**Supplementary Fig. 3f**), although it remains possible that regions outside the TRPV4 ARD can modulate RhoA interaction.”

5. The effect of TRPV4 on neurite outgrowth is puzzling. While previous studies reported that activation of TRPV4 promoted neurite outgrowth (Jang...Oh, JBC 2012) while other show retraction (Goswami PlosOne 2010), now McCray and cols. claim a negative effect of TRPV4.

The reviewer is correct to point out different reported effects of TRPV4 and TRPV4 channel activity on the cytoskeleton in the published literature. There are many possible reasons for these variations, including the vast complexities of cytoskeletal remodeling pathways among different cell types and within different subcellular structures analyzed (i.e. dendrites vs axons vs non-neuronal cell processes), as well as many specific aspects of the experimental paradigms employed. In particular, we believe that the time scale of evaluation of TRPV4 activity in various studies, i.e. minutes to hours vs. days, may be critical. Still, despite some conflicting results in the literature, we believe our work is consistent with the majority of published studies in relation to the effect of TRPV4 inhibition/activation and cytoskeletal changes. Specifically, our results are consistent with those from Goswami et al.⁴, who found that TRPV4 agonist stimulation over minutes to hours caused a rapid change in the actin cytoskeleton with retraction of filopodial processes, similar to what we have seen with neurite-like extension in MN-1 cells. Similarly, Jang et al.⁵ observed that over-expression of TRPV4 promoted neurite outgrowth, and siRNA-mediated knockdown of TRPV4 inhibited neurite outgrowth, results that are also in line with our own. In contrast to other studies and our work, Jang et al. reported promotion of neurite outgrowth with prolonged exposure to TRPV4 agonist (6 days). The results from this experimental paradigm, in which drugs were applied for many days, is distinct from the short-term exposures in our studies. Importantly, prolonged, long-term exposure to TRPV4 agonists can stimulate TRPV4 channel internalization and resultant reduced responsiveness as we and others have observed⁶.

6. Also in relation to these experiments, it may be better to use the Sholl index to quantify neurite outgrowth.

We appreciate the suggestion of using Sholl analysis for quantification of MN-1 neurite length. Indeed, we used this method to quantify the *Drosophila* sensory neuron dendrites (Fig. 6j). Unfortunately, the sheer number of MN-1 cells and their often overlapping protrusions prevented the use of this automated quantification method for these cells. While time-consuming, our method of blinded manual tracing produced highly reproducible, consistent, and robust neurite length quantification.

Reviewer #2:

Comments for McCray et al

This paper describes the novel interaction between TRPV4 channel and small GTPase RhoA. The authors found the interaction through the immunoprecipitation combined with

MASS spec, and further identified the interaction by NMR. Interestingly, the disease related mutation of TRPV4 reduced the interaction with RhoA. The interaction appears to be important for the neurite extension. These findings are interesting; however, it is not clear whether the interaction really happens in cells for the proposed function. I would like to propose some approaches to strengthen the authors findings.

We appreciate the reviewer's interest in our work and the insightful comments and suggestions. We have responded to each specific point below.

1. Both RhoA and TRPV4 are important for neuronal function, the synergy by the interaction should be examined. The authors performed NMR analysis, and thus there should be a lot of candidate residues to be mutated. Among them, the residues that are not reported for the dysfunction of TRPV4 might be included. Then it is possible to demonstrate that only the RhoA binding is changed by the mutation through the immunoprecipitation analysis combined with the Western blots or MASS spec, as authors performed for the identification of the RhoA as TRPV4 binding protein. These mutations in TRPV4 would further strengthen the importance of the interaction.

We appreciate the suggestion of undertaking a more detailed evaluation of specific residues in both TRPV4 and RhoA that mediate binding and functional interactions. In response to this insightful comment, we took a closer look at the NMR spectral intensity shifts in RhoA to identify specific RhoA residues that seemed to be particularly important for TRPV4 binding. We identified E47 and E54 as candidate residues, and generated RhoA GFP E47A and E54A expression vectors. Remarkably, RhoA E54A completely disrupted TRPV4 interaction by co-immunoprecipitation, while interaction was preserved with the E47A mutation (**Fig. 3f**). Both RhoA mutants were still able to interact normally with RhoGDI (**Supplementary Fig. 3g**), suggesting that they were properly folded and functional. These new data highlight the importance of the E54 residue in RhoA in mediating binding to TRPV4, likely via electrostatic interactions with ARD arginine residues. As we believe this new data is more specific and informative than the experiments showing disruption of TRPV4-RhoA interaction with C3 transferase treatment (previously Fig. 3f), we have omitted these data from this revised version of the manuscript. The new data regarding RhoA GFP E47A and E54A is discussed in the results as follows: "*We noted the largest decrease in signal intensity for the E54 residue in RhoA, and a smaller decrease in E47, suggesting these residues might be particularly important for TRPV4 binding. Indeed, introducing an E54A mutation into RhoA completely disrupted interaction with TRPV4, whereas interaction was preserved with the E47A mutant (Fig. 3f). Both mutants showed preserved interaction with RhoGDI, indicating that they were properly folded and functional (Supplementary Fig. 3g).*"

As our NMR data utilized isotope-labeled RhoA and not TRPV4, this analysis cannot be used to identify specific TRPV4 ARD residues that are involved in RhoA binding. We have attempted to obtain NMR spectra from TRPV4 ARD, but unfortunately the observed line broadening and consequent lack of sufficient spectral resolution precludes backbone amide assignments that are a requirement to pinpoint residue-specific RhoA binding effects. Instead, the original manuscript includes analysis of six different TRPV4 ARD mutations by co-immunoprecipitation (Fig. 2b-f, Supplementary Fig. 2a). In addition, we now include additional data examining mutations of the N terminus PIP2 binding site as well as the PRD, which disrupts interaction with PACSIN (**Supplementary Fig. 3f** and see Response to Reviewer 1 point #4). Together, our co-immunoprecipitation analysis of TRPV4 mutants clearly indicates that conserved arginines (R232, R237, R269, and R315) within the convex face of the ARD, and their

associated positive charge, are likely important for binding RhoA, as their mutation abolishes RhoA interaction. In contrast, neither mutations within the concave face of the ARD that cause skeletal dysplasia (I331F and D333G) nor disruption of the PIP2 binding domain or PRD disrupt RhoA binding. We believe these results provide important structural insights into how the TRPV4 ARD likely binds RhoA.

With regard to cytoskeletal remodeling and neurite outgrowth, we investigated the potential for synergistic effects of TRPV4 and RhoA by assessing neurite length in MN-1 cells (Fig. 6a-b, and 6e-h). We found that over-expression of RhoA inhibited the neurite outgrowth-promoting effect of TRPV4, and that mild pharmacologic inhibition of RhoA potentiated TRPV4-mediated neurite outgrowth. In addition, inhibition of RhoA rescued neurite outgrowth with expression of neuropathy mutant TRPV4, and also rescued dendritic degeneration in our fly model of TRPV4 neuropathy. These data suggest that TRPV4 and RhoA operate in a common pathway to control cytoskeletal changes and cellular process outgrowth. Our working model is that when WT TRPV4 channel is inactive, it binds and inhibits RhoA, which promotes neurite outgrowth, whereas TRPV4 channel activity leads to RhoA activation and neurite outgrowth inhibition (Figure 7).

2. Similar approach can be possible from RhoA. The RhoA mutations that might affect (only) the TRPV4 binding with lesser effect to the other binding proteins would be possible. But it would be difficult approach as compared to that from TRPV4 because of the abundance of RhoA binding proteins.

We have addressed this comment by analyzing TRPV4 interactions with RhoA E54A and E47A as described above.

3. The binding of Cdc42 was examined and shown to be negative. The binding of TRPV4 to Rac should also be examined.

We appreciate this suggestion as a way to help strengthen evidence for the specificity of TRPV4-RhoA interactions. We have now performed a co-immunoprecipitation of TRPV4-FLAG with epitope-tagged RhoA, Rac1, and Cdc42, and we show robust binding of TRPV4 and RhoA, but no binding of TRPV4 with Rac1 or Cdc42 (**Supplementary Fig. 1f**).

4. It is interesting that the TRPV4 binds to the GDP-bound form of RhoA. However, majority of GDP-bound RhoA should be in the cytoplasm. The co-localization of dominant-negative RhoA with TRPV4 should be shown. If it is not clear, then some complementary approaches, such as fluorescent lifetime and proximity ligation, and so on would be possible.

Our data is consistent with a model in which TRPV4 recruits GDP-bound RhoA to the plasma membrane. Like the reviewer, we also postulated that the localization of dominant negative RhoA might be informative and allow for a more definitive evaluation of TRPV4-RhoA co-localization and how TRPV4 affects RhoA subcellular distribution. However, we surprisingly found that dominant negative RhoA is relatively abundant at the plasma membrane when expressed alone (Reviewer Fig. 1a), a phenomenon that has been previously described⁷, and which essentially precludes it from being useful to probe interaction with TRPV4 at the membrane. In general, the largely diffuse cytosolic expression of RhoA makes quantitative

analysis of RhoA subcellular localization difficult (Fig 1d, Supplementary Fig. 1i, Supplementary Fig. 2c-d). We have attempted proximity ligation assays (PLA) of TRPV4 and interactors in the past, but did not find that it produced informative results. In particular, we have found that reliable signal with PLA requires highly specific antibodies as well as discreet subcellular localization of both proteins of interest, as diffuse localization (as is the case with RhoA) results in very high "proximity" signal that is not indicative of binding. As an example of the latter point, we found high PLA signal with epitope-tagged b PACSIN-1 and EGFP alone, which is not indicative of a true interaction (Reviewer Fig. 1b).

5. If TRPV4 inhibits RhoA upon TRPV4 activation, then it might be interesting mechanism for delaying the RhoA activation among Rho family GTPases. Then the time course of the activation of RhoA upon TRPV4 activation should be compared with the other small GTPases such as Cdc42 and Rac. RhoA might be simultaneously activated with Cdc42 and Rac in the absence of TRPV4 function.

We appreciate the suggestion of analyzing activation of other Rho GTPases in response to TRPV4 ion channel activity. To be clear, our model is that *inactive* TRPV4 binds *inactive* RhoA, but that TRPV4 ion channel activity leads to activation of RhoA that occurs rapidly after the influx of intracellular calcium. To investigate the relative contribution of TRPV4 ion channel activity for activation of other Rho GTPases, we have now performed activation assays of Rac1 and Cdc42 following TRPV4 agonist treatment (GSK101) under the identical paradigm as we used to demonstrate RhoA activation. In this system, activation of TRPV4 leads to a nearly 4-fold increase in RhoA activation, whereas there is no significant activation of Rac1, and a small but significant activation of Cdc42 (**Supplementary Fig. 5c-e**). The latter result is consistent with a recent publication that we now reference in the text as follows⁸: "*To test whether TRPV4 channel activity could regulate RhoA activation, we first demonstrated that the selective TRPV4 agonist GSK101 caused increased intracellular calcium in T-Rex-TRPV4 cells (Supplementary Fig. 5a) as well as phosphorylation of ERK (Supplementary Fig. 5b) as has been previously described⁹. Treatment of these cells with GSK101 also led to a robust increase in the active fraction of RhoA (Fig. 5a-b), but no activation Rac1 and a small but significant activation of Cdc42 (Supplementary Fig. 5c-e), consistent with a recent report⁸.*"

Reviewer #3:

The manuscript by McCray et al. investigates in a very detailed way the interactions

Reviewer Figure 1: (a) MN-1 cells were transfected with RhoA T19N-GFP (dominant negative) and imaged by confocal microscopy. RhoA T19N surprisingly shows enrichment at the plasma membrane, making it unsuitable as a marker for TRPV4-dependent membrane recruitment of RhoA. (b) MN-1 cells were transfected with GFP and PACSIN-1-Myc and analyzed by proximity ligation assay with antibodies to GFP and Myc. Note diffuse and robust proximity signal despite no physiologic interaction of the two proteins.

between TRPV4 and RhoA. TRPV4 interacts with the inactive, GDP-bound RhoA and the interaction is situated in the ankyrin repeat domain of TRPV4. This interaction is specifically disturbed by the presence of mutations in TRPV4 that induce neuropathies. RhoA binding to TRPV4 inhibits calcium influx, while influx of calcium activates RhoA. The model proposed by the authors is that TRPV4 serves a dual role. It can bind and inhibit RhoA when TRPV4 is in the closed state, while RhoA is activated once calcium is entering the cell upon stimulation of the TRPV4 channel. This bidirectional functional interaction is disturbed by neuropathy-inducing mutations. Last but not least, it is shown that the TRPV4-RhoA interaction regulates cell morphology which is disrupted by the neuropathy mutations. Inhibition of RhoA can restore neurite length in cells and in *Drosophila* larvae and RhoA inhibition is proposed as a potential therapeutic strategy. Overall, this manuscript contains an overwhelming amount of novel data and convincingly links RhoA to TRPV4 and provides a potential explanation for the effect of neuropathy-inducing mutations in TRPV4.

Detailed information is provided on every aspect of this study and all statistical assays are correct. This manuscript constitutes a very relevant addition to what is currently known about the potential mechanism behind the TRPV4 mutations causing peripheral neuropathies. I have a number of remarks but none of them questions the relevance of this work.

We thank the reviewer for the very positive feedback regarding the novel findings of our work and their implications for understanding the pathogenesis of TRPV4 neuropathy. We have addressed the reviewer's comments below.

Major remarks

1. The rationale for selecting RhoA from the mass spec data is not completely clear. There are many more hits present and in some cases the difference between WT and mutant is higher than for RhoA. Moreover, the published yeast-two hybrid screen also detects many more interactors. Not so clear why exactly RhoA was selected and the question arises whether these other hits could also play a role in the disease process.

We appreciate this suggestion and have now included rationale for focusing on RhoA in the revised text as follows: "*We chose to focus subsequent studies on the small GTPase RhoA as it demonstrated high peptide counts, appeared enriched with WT TRPV4, and plays a well-known role in regulating the actin cytoskeleton.*" Additional hits from our screen are being actively explored.

2. The impression exists (and it is also mentioned in the discussion) that there is a much closer connection between the observations recently published by the same research group in Nature Communications and the ones reported in this manuscript. The link between CamKII and RhoA could eventually be investigated in more detail using the fly models that are available (which could be a follow-up study).

We agree that there might be a very interesting relationship between the dramatic effects of CaMKII knockdown/inhibition on TRPV4 calcium influx and toxicity that we reported in our recent manuscript¹⁰ and the functional effects of RhoA reported in the current manuscript. We hypothesize that CaMKII may be a critical downstream node following TRPV4 channel activity,

and that it may be directly or indirectly responsible for RhoA activation through modulation of specific GEFs. Such studies will be the focus of future work both *in vitro* and using the fly model. In response to the reviewer comment, we have expanded the discussion of how CaMKII and RhoA may work in concert to respond to TRPV4-mediated calcium influx: “*Calcium-mediated activation of RhoA is thought to involve phosphorylation of RhoA regulatory proteins by calcium-sensitive kinases such as CaMKII. This is particularly intriguing given our recent work showing that CaMKII inhibition is protective in Drosophila and cultured mammalian neurons expressing TRPV4 neuropathy mutants. Together, these data suggest that CaMKII may be a critical downstream node that transduces TRPV4-mediated calcium influx to regulate RhoA activity and cytoskeletal changes.*”

3. The interaction between RhoA and TRPV4 is always shown in overexpression systems (or in cell free systems). A crucial question is whether similar interactions also occur in cells when both TRPV4 and RhoA are expressed at physiological levels. If this is technically impossible to investigate, this important limitation should be discussed in more detail.

The reviewer is correct to point out that all co-immunoprecipitation experiments in the original manuscript involve some degree of exogenous expression of either RhoA and/or TRPV4. The reviewer is also correct in the assumption that experiments examining endogenous TRPV4 are technically quite challenging, as is well-documented in the literature. To our knowledge, there are very few examples of demonstration of protein-protein interactions with endogenous TRPV4 rather than with epitope-tagged TRPV4. Even with PACSINs, perhaps the most well-validated TRPV4 interactors, there is extremely limited data demonstrating co-immunoprecipitation of endogenous TRPV4 and PACSIN¹¹. The primary reasons for the difficulties in working with endogenous TRPV4 are the lack of highly specific commercially available antibodies and the low abundance of TRPV4 in most cell types and tissues. Immunoprecipitation of RhoA rather than TRPV4 could help circumvent the problem of TRPV4 antibody quality, but we were unfortunately unable to find a suitable antibody to efficiently immunoprecipitate endogenous RhoA (despite trying several antibodies used for immunoprecipitation in published studies). However, after testing multiple TRPV4 antibodies by western blot, immunofluorescence, and immunoprecipitation, we have found acceptable results with a polyclonal rabbit antibody from Abcam (ab39260). Even with this antibody, the fact remains that physiological levels of TRPV4 in most cells and tissues are so low such that many biochemistry approaches, including immunoprecipitation, are not feasible. After testing many cell and tissue types and extensive optimization of experimental conditions, we found that mouse choroid plexus demonstrates sufficient TRPV4 expression such that immunoprecipitation of TRPV4 can be performed efficiently. We have also validated that TRPV4 is expressed at the cell surface in primary cultures of choroid epithelial cells. Using freshly dissected choroid plexus, we have now successfully performed immunoprecipitation of endogenous TRPV4 and demonstrated co-immunoprecipitation of endogenous RhoA (**Supplementary Fig. 1f**).

We also would like to point out that, whenever possible, we have validated our findings regarding TRPV4-RhoA interaction by demonstrating co-immunoprecipitation of *endogenous* RhoA and induced TRPV4-FLAG from T-Rex TRPV4 cells (Fig. 1c, Fig. 2f, **Fig 5i**, Supplementary Fig. 1i, and **Supplementary Fig. 5g**). While TRPV4 is technically overexpressed in this system, we use a relatively low level of induction in all experiments such

that total TRPV4 levels are only modestly increased above endogenous levels (**Supplementary Fig. 1e**). Together with endogenous co-immunoprecipitation of choroid plexus, we believe these results strongly argue for interaction of TRPV4 and RhoA within cells under physiological conditions.

4. The fly data are limited to the investigation of the impact of a dominant negative overexpression of Rho1 on the morphology of sensory neurons in the fly larvae. It would be interesting to know whether this also affects the other phenotypes observed in the mutant TRPV4 flies.

We thank the reviewer for this suggestion. In response, we examined the effect of Rho1 dominant negative on additional phenotypes in TRPV4 R269C-expressing flies that we described in Woolums et al., Nat Commun 2020¹⁰, including sensory neuronal axonal projection degeneration within the ventral nerve cord and failure of wing opening. We present new data showing that expression of Rho1 dominant negative leads to dramatic rescue of ventral nerve cord axonal projections (**Fig. 6k-l**) and significantly improves the failure of wing opening in mutant flies (**Fig. 6m**). These data further strengthen the finding that Rho inhibition can reduce downstream toxicity of neuropathy mutant TRPV4. These data are now referenced in the results as follows: “*Strikingly, we found that genetic inhibition of the activity of the Drosophila RhoA ortholog Rho1 by expression of dominant negative Rho1[T19N] dramatically rescued both dendritic (Fig. 6i-j) and axonal (Fig. 6k-l) degeneration with expression of neuropathy mutant TRPV4. We also found strong rescue of wing opening failure with expression of Rho1[T19N] (Fig. 6m).*”

Minor remarks

- If possible, MW markers should be indicated next to the Western blots.

Molecular weight markers have now been added to all western blot images.

- Fig. 1a,b What is the explanation for the fact that the input is so different for the experimental condition that is similar on both blots? It is also remarkable that the intensity of the GFP band is so low in panel a, while the FLAG band in panel b is so intense. This doesn't seem to correlate with the input.

The different intensities of the bands in question were due to inconsistencies in exposure times. To show the robustness of the co-immunoprecipitation more clearly, we have substituted the blot shown in Figure 1a for a more recent blot.

- In COS7 cells (Suppl. Fig. 1g) RhoA seems to be enriched in the nucleus (or at least a perinuclear region). Which is the explanation for this?

We thank the reviewer for bringing the nuclear signal to our attention. With respect to the perinuclear GFP signal, this likely reflects RhoA localization to the Golgi, which has been described⁷. Regarding the nuclear signal, we reviewed many images of RhoA-GFP in various cell types, and we see that this phenomenon is unique to COS7 cells. We suspect this is due to low-level cleavage of the GFP tag from RhoA (which we sometimes see by western blot), and

subsequent nuclear translocation of free GFP. Despite this unexpected phenomenon, we believe that the majority of the GFP signal in these cells is representative of RhoA-GFP distribution as the fluorescence pattern is clearly distinct from what is seen with expression of free GFP. As such, we believe that these images are still informative with respect to subcellular localization of RhoA and co-localization with TRPV4.

- Fig. 1g: The effect of EDTA is difficult to judge as the condition without EDTA is missing. In the absence of this control, it is also difficult to judge whether the EDTA + excess Mg²⁺ condition leads to more binding.

Based on this comment, we realized that the text in question was confusing and somewhat misleading. To be more precise, we have modified the text to describe that Figure 1g shows that excess magnesium can counteract the effect of EDTA in the buffer. The modified text now reads as follows: "*We found that TRPV4-RhoA interaction was markedly enhanced by addition of magnesium to buffers that contain EDTA (Fig. 1g), suggesting that efficient TRPV4-RhoA interaction depends on magnesium-dependent nucleotide binding.*" We also changed the figure legend to read as follows: "*(g) Co-immunoprecipitation of MN-1 cells transfected with TRPV4-FLAG and RhoA-GFP in the presence of EDTA (1 mM in all conditions), which disrupts RhoA nucleotide binding, with or without excess magnesium (10 mM), which stabilizes RhoA nucleotide binding, demonstrates that TRPV4-RhoA interactions are nucleotide-dependent.*"

- Only the intensity of the blot shown in Fig. 1j is quantified. Not clear why that is not systematically done with all the IP blots on this Figure. The number of blots quantified should also be indicated in the legend.

We chose not to quantify the band intensities from Figure 1h as the main point was the complete absence of binding of Q63L RhoA rather than the more subtle differences between WT RhoA and T19N RhoA. On the other hand, the effect of GDP and GTP γ S is more granular and becomes more apparent with quantification. The number of experiments/blots analyzed in Figure 1j is n=4 and is indicated in the figure legend.

- Not so clear whether the results using the TRPV4 antagonist are indeed shown on Fig. 2.

We thank the reviewer for pointing out a lack of clarity in how the data was originally described, and we have now modified the text accordingly. The co-immunoprecipitation blot showing the interaction of RhoA and TRPV4 in the absence of antagonist is presented in the supplemental data (Supplementary Fig. 2a), while co-immunoprecipitation blots and quantification in the presence of antagonist or the ion channel pore inactivating mutation (M680K) are presented in Figure 2b-e. The text describing these data has now been modified to read as follows: "*we performed TRPV4-RhoA co-immunoprecipitation experiments in the presence (Fig. 2b-e) or absence (Supplementary Fig. 2a) of TRPV4 ion channel inhibition.*"

- Text on Fig. 3e is difficult to read.

The important amino acid residues have been reformatted to increase legibility.

- The transfection efficiency of TRPV4 in the calcium experiment seems to be rather low. Moreover, the impression exists that also calcium increases are observed in cells that are TRPV4 negative. Is this a technical issue?

We appreciate the reviewer calling our attention to this problem. The contrast on the images of GFP-TRPV4 was very low and has now been adjusted to more accurately demonstrate TRPV4 expression. There are only rare non-transfected cells that respond to hypotonic saline, and GFP transfected cells on average show no response to hypotonic saline as shown in Supplementary Figure 4a.

- If possible, the supplementary Figure 7 should be integrated in the manuscript as it is very helpful to understand the conclusions of this paper. In line with my major remark and if possible, CamKII should also be integrated in this figure.

We agree with this suggestion and have now placed the schematic in the main text as Figure 7. As we do not yet have any data connecting RhoA and CaMKII in a single pathway, we have not included CaMKII in the schematic.

- The discussion is a bit too long and should focus more on the most important aspects of this study

We appreciate this suggestion. We have now shortened the discussion and removed some sections that we deemed less essential to the main findings of the manuscript (reduced from ~1700 words to ~1400 words)

- Scale bars are missing on the pictures present in the supplementary material.

We have added scale bars as necessary.

- Ref 26 is incomplete.

The reference manuscript was still in the editorial process when this manuscript was submitted, but we have now included the full reference.

- 1 Garcia-Elias, A. *et al.* Phosphatidylinositol-4,5-biphosphate-dependent rearrangement of TRPV4 cytosolic tails enables channel activation by physiological stimuli. *Proc Natl Acad Sci U S A* **110**, 9553-9558, doi:10.1073/pnas.1220231110 (2013).
- 2 Goretzki, B. *et al.* Structural Basis of TRPV4 N Terminus Interaction with Syndapin/PACSIN1-3 and PIP2. *Structure* **26**, 1583-1593 e1585, doi:10.1016/j.str.2018.08.002 (2018).
- 3 Cuajungco, M. P. *et al.* PACSINs bind to the TRPV4 cation channel. PACSIN 3 modulates the subcellular localization of TRPV4. *J Biol Chem* **281**, 18753-18762, doi:10.1074/jbc.M602452200 (2006).
- 4 Goswami, C., Kuhn, J., Heppenstall, P. A. & Hucho, T. Importance of non-selective cation channel TRPV4 interaction with cytoskeleton and their reciprocal regulations in cultured cells. *PLoS One* **5**, e11654, doi:10.1371/journal.pone.0011654 (2010).
- 5 Jang, Y. *et al.* Axonal neuropathy-associated TRPV4 regulates neurotrophic factor-derived axonal growth. *J Biol Chem* **287**, 6014-6024, doi:10.1074/jbc.M111.316315 (2012).
- 6 Jin, M. *et al.* Determinants of TRPV4 activity following selective activation by small molecule agonist GSK1016790A. *PLoS One* **6**, e16713, doi:10.1371/journal.pone.0016713 (2011).
- 7 Michaelson, D. *et al.* Differential localization of Rho GTPases in live cells: regulation by hypervariable regions and RhoGDI binding. *J Cell Biol* **152**, 111-126, doi:10.1083/jcb.152.1.111 (2001).
- 8 Yang, W. *et al.* TRPV4 activates the Cdc42/N-wasp pathway to promote glioblastoma invasion by altering cellular protrusions. *Sci Rep* **10**, 14151, doi:10.1038/s41598-020-70822-4 (2020).
- 9 Chen, Y. *et al.* Transient Receptor Potential Vanilloid 4 Ion Channel Functions as a Pruriceptor in Epidermal Keratinocytes to Evoke Histaminergic Itch. *J Biol Chem* **291**, 10252-10262, doi:10.1074/jbc.M116.716464 (2016).
- 10 Woolums, B. M. *et al.* TRPV4 disrupts mitochondrial transport and causes axonal degeneration via a CaMKII-dependent elevation of intracellular Ca²⁺. *Nat Commun* **11**, 2679, doi:10.1038/s41467-020-16411-5 (2020).
- 11 D'Hoedt, D. *et al.* Stimulus-specific modulation of the cation channel TRPV4 by PACSIN 3. *J Biol Chem* **283**, 6272-6280, doi:10.1074/jbc.M706386200 (2008).

Reviewers' Comments:

Reviewer #1:

Remarks to the Author:

Authors have correctly addressed all questions.

Reviewer #2:

Remarks to the Author:

The manuscript was significantly improved and should be accepted for publication.

Reviewer #3:

Remarks to the Author:

I have no further comments or remarks.

Reviewer #4:

Remarks to the Author:

In this manuscript, McCray et al use a mass-spectrometry screen to identify binding partners of the TRPV4 channel, and use biochemical methods to follow up this screen, establishing RhoA as a potential binding partner and channel regulator. As a mass spectrometry expert I do not have familiarity with these pathways, so do not feel confident to comment on the biological relevance and context of these experiments (it seems the other reviewers have already done this in detail), but I do have some general minor comments on the presentation of the data and the initial mass spectrometry findings. Only one of these issues is major.

Major comments

1) All IP western blots are very heavily cropped. When I tried to look at the uncropped versions in the submission, I became very confused. For 1A and 1B GFP the blots don't seem to match. This could be extremely different exposures, but the major bands look a different shape to me too. 1A and 1B may have been swapped, it is very difficult to tell. The full blots need re-examining, much better labelling, and if possible, the actual size ladder should be included on these blots, not just the size markers added by the authors. In 1e the uncropped blots are presented the opposite way round to the main figure. Was this file perhaps not updated after review? Unfortunately seeing how unclear this data is gives me concerns about the reliability of this paper.

Minor comments

1) The experiments are dense and to the general reader it is difficult to understand how one set of IPs is distinguished from the preceding experiments – a rudimentary cartoon at the beginning of each main figure would be really helpful for context of each experiment.

2) The wash buffer used for the mass-spectrometry IPs should be detailed

3) The quantification of binding proteins by spectral counting has mostly been replaced by MS1 peak intensity measurements in the proteomics field. I am presuming given the amount of work that followed this screen that the MS was done some years ago, so this is not surprising. There are a number of issues with spectral counting that affect accurate quantification. I don't think this is a huge issue given the quants are only used to prioritize the follow up list, but it may be good to mention a couple of caveats about this technique in the text.

4) I presume Table S1 is heavily filtered, as it shows none of the normal contaminants that are usually seen in IP experiments. The methods section should detail how this filtering was performed, and consider sharing the more "raw" version of the output data.

5) Are the raw files for the proteomics experiment published uploaded to a data repository? This should be done before publication if not.

6) In the Figure 2 legend parts b and c are incorrectly referred to.

7) There doesn't seem to be a signal from IgG in the IP samples. This makes sense for the mass spectrometry where elution was performed with excess FLAG, but for the western blots I would probably expect elution of at least some IgG from the beads with use of Laemmli and BME. Was heat also used to elute? Were the antibodies crosslinked to the beads for IP? Given endogenous Rho is the same size as the heavy chain this is important information for reproducibility.

Becky Carlyle

Instructor, MGH Department of Neurology

Thank you for the re-review of our manuscript. We appreciate the original reviewers' statements that the manuscript is significantly improved and now suitable for publication. We also thank Reviewer 4 for providing specific input as requested by the editorial staff. We have addressed all of the concerns raised by Reviewer 4 in the point by point response below. The only change to the appearance of the figures is an additional schematic to clarify a specific experiment (Supplementary Figure 1j). As no additional experimental work was suggested or requested, the data in this revision are unchanged. We apologize for the confusion created by inadequate labeling of our uncropped western blot images in the Source Data file as well as the inadvertent switch of the uncropped images of the blots presented in Figure 1a and 1b. We thank Reviewer 4 for spotting these errors, and we have rectified these in the current submission. We look forward to the editorial response to the revised manuscript.

Reviewer 1:

Authors have correctly addressed all questions.

We appreciate the reviewer's help in improving the manuscript.

Reviewer 2:

The manuscript was significantly improved and should be accepted for publication.

We appreciate the reviewer's help in improving the manuscript and recommendation that this work be published.

Reviewer 3:

I have no further comments or remarks.

We appreciate the reviewer's help in improving the manuscript.

Reviewer 4:

In this manuscript, McCray et al use a mass-spectrometry screen to identify binding partners of the TRPV4 channel, and use biochemical methods to follow up this screen, establishing RhoA as a potential binding partner and channel regulator. As a mass spectrometry expert I do not have familiarity with these pathways, so do not feel confident to comment on the biological relevance and context of these experiments (it seems the other reviewers have already done this in detail), but I do have some general minor comments on the presentation of the data and the initial mass spectrometry findings. Only one of these issues is major.

Major comments

1) All IP western blots are very heavily cropped. When I tried to look at the uncropped versions in the submission, I became very confused. For 1A and 1B GFP the blots don't seem to match. This could be extremely different exposures, but the major bands look a different shape to me too. 1A and 1B may have been swapped, it is very difficult to tell. The full blots need re-examining, much better labelling, and if possible, the actual size ladder should be included on these blots, not just the size markers added by the authors. In 1e the uncropped blots are presented the opposite way round to the main figure. Was

this file perhaps no updated after review? Unfortunately seeing how unclear this data is gives me concerns about the reliability of this paper.

The reviewer is correct to point out that the western blots in the main figures are cropped to include only the relevant portions of the blots. The full, uncropped images have been provided as a “Source Data” file, which allows the reader to assess the quality of the blots and specificity of antibodies used. As the reviewer correctly surmised, the uncropped blots for Figure 1a and Figure 1b were accidentally switched in the Source Data file. We apologize for this error, which has now been corrected. In the original Source Data file, we did not extensively label the blots as we felt this was largely duplicative of the labeling that is already in the figures and supplementary figures. However, as this has created confusion, we have more fully labeled all of the Source Data images, including labeling of all wells, band identities, antibodies used, cropped images of the molecular weight markers, and when indicated, the nature of the samples loaded in gel (e.g. inputs versus immunoprecipitations versus Rho activation).

Regarding the comment that “1e the uncropped blots are presented the opposite way round to the main figure,” we believe the reviewer is referring to Figure 1g rather than Figure 1e. For the Source Data file related to Figure 1g, the IP blot was placed on the left and the input blot on the right, whereas the image for Figure 1g has the input blot on the left and the IP blot on the right. We apologize for this error, which has now been corrected in the Source Data file.

We feel that the reviewer comment that “seeing how unclear this data is gives me concerns about the reliability of this paper” warrants a specific rebuttal. We believe this comment is overly critical, as it appears that the basis of the comment is the inadvertent switching of the images of two figure panels in the Source Data file as discussed above. Reviewer 4 nor the other three reviewers have raised significant methodological, technical, or interpretive errors or inconsistencies that would warrant questioning the reliability of this body of work. In support of the reliability and reproducibility of the data in the manuscript, we note that the experiment presented in Figure 1a (demonstration of co-immunoprecipitation of WT TRPV4-FLAG with RhoA-GFP but not GFP alone) is essentially presented in the manuscript eight times (Figure 1a, Figure 1e, Figure 1g, Figure 2b, Figure 2c, Figure 3f, Supplementary Figure 2a, and Supplementary Figure 3f) with the same robust results.

Minor comments

1) The experiments are dense and to the general reader it is difficult to understand how one set of IPs is distinguished from the preceding experiments – a rudimentary cartoon at the beginning of each main figure would be really helpful for context of each experiment.

With the exception of the co-immunoprecipitation experiments using immunopurified TRPV4-FLAG and RhoA-GFP (Figure 1e and Supplementary Figure 1j), the experimental approaches used throughout the paper are straightforward and consistent with routine approaches used in molecular and cellular biology. We have now included a schematic (Supplementary Figure 1j) for the co-incubation experiment in Figure 1e. We believe that the other experimental approaches are adequately described in the text and methods section such that the average reader of *Nature Communications* will have no difficulty in understanding the design, purpose, and conclusions of the experiments presented.

2) The wash buffer used for the mass-spectrometry IPs should be detailed

We thank the reviewer for pointing out this omission. The components of the wash buffer have now been added to the methods.

3) The quantification of binding proteins by spectral counting has mostly been replaced by MS1 peak intensity measurements in the proteomics field. I am presuming given the amount of work that followed this screen that the MS was done some years ago, so this is not surprising. There are a number of issues with spectral counting that affect accurate quantification. I don't think this is a huge issue given the quants are only used to prioritize the follow up list, but it may be good to mention a couple of caveats about this technique in the text.

We thank the reviewer for this expert comment. However, we did not feel that a detailed discussion of the merits and limitations of spectral counting versus other approaches is appropriate for the main text. Instead, we have added the following to the end of the "Liquid chromatography-mass spectrometry" section of the methods: "*Prioritization of potential TRPV4 binding proteins was in part based on spectral counts as described in Supplementary Table 1. However, due to inherent limitations with quantification based on spectral counts, this approach has now largely been replaced by quantification based on precursor signal intensity, such as MS1 intensity measurements.*"

4) I presume Table S1 is heavily filtered, as it shows none of the normal contaminants that are usually seen in IP experiments. The methods section should detail how this filtering was performed, and consider sharing the more "raw" version of the output data.

The reviewer is correct that we applied multiple filters to arrive at the list presented in Table S1. This process is described generally in the main text as follows: "Stratification of potential interactors based on relative enrichment in WT TRPV4 versus R237L TRPV4 and negative control (empty vector transfection)." We have now included additional detail of the filtering strategy in the legend for Supplementary Table 1: "*Results were filtered by setting a minimum threshold of 20 spectral counts and removing proteins that had less than 2-fold spectral count enrichment in WT TRPV4 as compared to the empty vector control. Further filtering to identify interactions that are potentially disrupted by the R237L neuropathy mutation was performed by removing proteins that had less than 1.5-fold spectral count enrichment in WT TRPV4 as compared to R237L TRPV4.*"

5) Are the raw files for the proteomics experiment published uploaded to a data repository? This should be done before publication if not.

We thank the reviewer for this suggestion. We are in the process of uploading the raw proteomics data to the PRIDE database, and it will also be submitted to the ProteomeXchange Consortium repository prior to publication of the manuscript.

6) In the Figure 2 legend parts b and c are incorrectly referred to.

We appreciate the reviewer noting this error, which we have now corrected.

7) There doesn't seem to be a signal from IgG in the IP samples. This makes sense for the mass spectrometry where elution was performed with excess FLAG, but for the western blots I would probably expect elution of at least some IgG from the beads with use of Laemmli and BME. Was heat also used to elute? Were the antibodies crosslinked to the beads for IP? Given endogenous Rho is the same size as the heavy chain this is important information for reproducibility.

As described in the methods, co-immunoprecipitation samples were eluted from beads using Laemmli sample buffer with β -mercaptoethanol. Samples were also heated, which was not originally described in the methods. Thus, we have edited the methods to read as follows: "*To elute bound proteins, Laemmli sample buffer with β -mercaptoethanol was added to the beads and samples were heated for 10 min at 70°C.*" As this manuscript relies heavily on immunoprecipitation and western blotting approaches, we went to great lengths to identify suitable reagents and optimize experimental conditions for these experiments. For co-immunoprecipitation experiments, we used different antibody species for immunoprecipitation and western blotting whenever possible (usually employing mouse monoclonal antibodies for immunoprecipitation and rabbit polyclonal antibodies for western blotting) to minimize the appearance of IgG bands in our blots. We also used light chain-specific HRP-conjugated secondary antibodies to avoid the appearance of IgG heavy chain on western blots. Still, there were some cases in which IgG light chain does appear in the blots (Figure 1b, Supplementary Figure S1j, Figure 1g, and Supplementary Figure 2b). We did not perform crosslinking for any experiments in this manuscript.

Becky Carlyle
Instructor, MGH Department of Neurology

Reviewers' Comments:

Reviewer #4:

Remarks to the Author:

I am very happy that all my major concerns have been addressed - as the supplementary blots have now been reordered and replaced with the appropriate images, it is much easier to understand what was happening. The extra labeling is of course, now somewhat redundant.

Two minor things that are not essential to add at this point and I will leave up to the authors, but I think would help with general audience understanding. In my point about adding a cartoon, I was in fact referring to a figure more like Figure 7, not a basic explanation of an IP, which of course is familiar to readers of Nature Communications. As a general reader, to suddenly localize yourself in a pathway you are not familiar with, and the subtleties of interactions within a pathway is tricky. It is somewhat unconventional to put this figure up front, but it would help with piecing apart which part of the complete picture each following figure specifically deals with.

Second - it would be great to specifically mention the IP optimizations, such as avoiding same species antibodies and using the light chain specific antibody to visualize IP results in the IP methods section. I missed this the first time as it was only mentioned in the long list of antibodies used across the full paper.

Thank you for the additional review of our manuscript. We are pleased that reviewer #4 feels that all major concerns have been addressed. We have addressed the minor concerns raised by Reviewer #4 as described below. As no additional experimental work was suggested or requested, the data in this revision are unchanged. We look forward to publication of the revised manuscript.

Reviewer #4:

1) I am very happy that all my major concerns have been addressed - as the supplementary blots have now been reordered and replaced with the appropriate images, it is much easier to understand what was happening. The extra labeling is of course, now somewhat redundant.

We are pleased that the reviewer now finds that all major concerns have been addressed and that the data presentation is easier to understand.

2) Two minor things that are not essential to add at this point and I will leave up to the authors, but I think would help with general audience understanding. In my point about adding a cartoon, I was in fact referring to a figure more like Figure 7, not a basic explanation of an IP, which of course is familiar to readers of Nature Communications. As a general reader, to suddenly localize yourself in a pathway you are not familiar with, and the subtleties of interactions within a pathway is tricky. It is somewhat unconventional to put this figure up front, but it would help with piecing apart which part of the complete picture each following figure specifically deals with.

We appreciate the reviewer's thoughts regarding how to improve clarity of presentation of the data. We feel that the schematic presented in Figure 7 incorporates the majority of the important findings and conclusions of the paper, and as such, an additional schematic would not necessarily improve the manuscript. However, to more clearly guide the reader through which figures correspond to which specific elements of the Figure 7 schematic, we have added references to specific figures throughout the figure legend as follows: *"Schematic representation of how TRPV4 neuropathy mutations disrupt normal TRPV4-RhoA functional interactions. TRPV4 binding to RhoA (Figure 1) is disrupted by TRPV4 neuropathy mutations (Figures 2 and 3). Impaired binding causes loss of TRPV4-mediated RhoA inhibition (Figure 4) and disruption of RhoA-dependent TRPV4 ion channel inhibition (Figure 4). Excessive calcium influx via neuropathy mutant TRPV4 causes further activation of RhoA (Figure 5). Together, disrupted TRPV4-RhoA interactions lead to increased TRPV4 ion channel activity, increased RhoA activation, RhoA-mediated actomyosin contraction, actin stress fiber formation, and cell process retraction (Figures 5 and 6)."*

Second - it would be great to specifically mention the IP optimizations, such as avoiding same species antibodies and using the light chain specific antibody to visualize IP results in the IP methods section. I missed this the first time as it was only mentioned in the long list of antibodies used across the full paper.

We appreciate the reviewer's suggestion to highlight these importance methodological details. We have now added the following to the Co-immunoprecipitation section of the Methods: "To

reduce the appearance of IgG heavy chain and light in western blots, different antibody species were used for immunoprecipitation and western blotting whenever possible. In addition, the appearance of IgG heavy chain was reduced by using IgG light chain-specific HRP-conjugated secondary antibodies.”